# Closed-form Solutions: A New Perspective on Solving Differential Equations

**Shu Wei** [1 2]   **Yanjie Li** [1]   **Lina Yu** [1 2]   **Weijun Li** [1 3 4]   **Min Wu** [1]   **Linjun Sun** [1]   **Jingyi Liu** [1]   **Hong Qin** [1 3]
**Yusong Deng** [1 2]   **Jufeng Han** [1 3]   **Yan Pang** [1 4]

## Abstract

The quest for analytical solutions to differential equations has traditionally been constrained by the need for extensive mathematical expertise. Machine learning methods like genetic algorithms have shown promise in this domain, but are hindered by significant computational time and the complexity of their derived solutions. This paper introduces **SSDE** (Symbolic Solver for Differential Equations), a novel reinforcement learning-based approach that derives symbolic closed-form solutions for various differential equations. Evaluations across a diverse set of ordinary and partial differential equations demonstrate that SSDE outperforms existing machine learning methods, delivering superior accuracy and efficiency in obtaining analytical solutions.

## 1. Introduction

Differential equations (DEs) are foundational to mathematics, physics, and the natural sciences, providing abstract models for diverse physical phenomena. For example, Poisson's equation governs the distribution of electrostatic potential, while the heat equation describes temperature variations within an object (Evans, 2022). Solving these DEs is critical for understanding and predicting the dynamics of complex physical systems (Alkhadhr & Almekkawy, 2021; Savović et al., 2023).

Deriving analytical solutions requires rigorous analysis of their existence, uniqueness, stability, and behavioral properties, often leveraging advanced mathematical theories and computational techniques (Beck, 2012; Friz et al., 2020; Aderyani et al., 2022). For example, solving linear partial differential equations (PDEs) via the superposition principle involves determining Green's functions, $G(\mathbf{x}, \mathbf{x}')$, which represent solutions to PDEs with a point source at $\mathbf{x}'$. This process demands deep insights into the properties of Green's functions, especially under complex boundary conditions or in multidimensional domains (Duffy, 2015). Nonlinear PDEs, however, pose greater challenges due to their inherent complexity, rendering analytic solutions intractable and necessitating numerical methods. Traditional numerical methods, such as the finite volume method, are prone to discretization errors and rely heavily on meshing, which can lead to convergence issues, especially with finely granulated meshes (Moukalled et al., 2016; Li et al., 2023b). Machine learning (ML) methods have emerged as powerful tools to address these limitations, significantly enhancing the ability to model and solve complex PDEs.

Physics-Informed Neural Networks (PINNs) (Raissi et al., 2019; Cuomo et al., 2022), grounded in the universal approximation theorem (Hornik et al., 1989), effectively approximate solutions to diverse physical systems. However, due to the nature of neural networks, these methods often exhibit low computational efficiency and struggle to accurately satisfy physical constraints, resulting in compromised convergence accuracy, particularly for stiff or high-frequency problems (Wang et al., 2021; Rao et al., 2023). The advent of neural operator methods, such as Deep Operator Network (DeepONet) (Lu et al., 2021a) and the Fourier Neural Operator (FNO) (Li et al., 2023b), which learn mappings between functions, offers a promising avenue to tackle these issues. Yet, these neural operator algorithms typically require extensive labeled data for training. Moreover, both traditional numerical methods and deep learning approaches sacrifice the interpretability of analytical solutions. This lack of interpretability poses challenges in accurately grasping system dynamics and limits the ability to extrapolate beyond the computational domain, thereby constraining their applicability in scientific research and engineering applications.

Recent advances in symbolic regression have enabled the derivation of interpretable solutions to DEs without requiring expert mathematical knowledge, through data-driven approaches. Genetic programming (GP)-based approaches

---

[1]AnnLab, Institute of Semiconductors, Chinese Academy of Sciences, Beijing, China [2]College of Materials Science and Opto-Electronic Technology, University of Chinese Academy of Sciences, Beijing, China [3]School of Integrated Circuits, University of Chinese Academy of Sciences, Beijing, China [4]School of Industry-education Integration, University of Chinese Academy of Sciences, Beijing, China. Correspondence to: Lina Yu <yulina@semi.ac.cn>, Min Wu <wumin@semi.ac.cn>.

*Proceedings of the $42^{nd}$ International Conference on Machine Learning*, Vancouver, Canada. PMLR 267, 2025. Copyright 2025 by the author(s).

employ physics-regularized fitness functions to evolve symbolic solutions. Tsoulos et al. (2006) propose a GP method that evolves solution expression genomes modeled using Formal Grammars (HOPCROFT, 1979). Oh et al. (2023) introduce a customizable GP approach (Randall et al., 2022) that directly evolves expression trees as solutions for this task. Cao et al. (2023; 2024) enhance GP algorithms with pruning techniques and transfer learning. Similarly, Kamali et al. (2015) and Boudouaoui et al. (2020) develop population-based methods that emulate heuristic search behaviors inspired by resource-seeking populations. However, these approaches often yield complex solutions with limited interpretability and incur high computational costs, particularly in high-dimensional problems, where finding accurate solutions remains challenging. In parallel, alternative paradigms have emerged. Liang and Yang propose the FEX method (2022), which uses reinforcement learning to search for weighted symbolic networks to solve differential equations. Liu's Kolmogorov-Arnold Networks (2025) can be used to derive symbolic solutions through network symbolization. While both methods improve fitting performance by introducing additional parameters, they frequently lack interpretability. Majumdar et al. propose an approach utilizing trained PINNs to generate datasets, subsequently applying symbolic regression to extract expressions (2023). This method falls short of adhering to physical constraints, primarily due to inherent limitations in numerical precision and error propagation. The core challenge lies in balancing three critical aspects: (1) maintaining strict adherence to physical laws, (2) ensuring computational efficiency, and (3) preserving human-interpretable symbolic forms. Current symbolic regression approaches either sacrifice interpretability for accuracy or face scalability issues in complex problems. Alternative paradigms, while offering improved fitting capabilities, fundamentally differ from our goal of deriving precise, physics-compliant symbolic solutions. To overcome these limitations, we propose a novel reinforcement learning (RL)-based method for deriving closed-form symbolic solutions to DEs. Our contributions are as follows:

- The introduction of the Symbolic Solver for Differential Equations (**SSDE**), an RL-based paradigm that directly derives closed-form symbolic solutions to differential equations with high interpretability.

- The development of the **Risk-Seeking Constant Optimization (RSCO)** algorithm, which accelerates constant optimization while preserving accuracy and convergence.

- A novel form of symbolic solutions expressed as parametric expressions, enhancing the applicability of SSDE for discovering symbolic solutions to DEs in any single dimension.

- An innovative exploration technique that enables SSDE to recursively solve high-dimensional PDEs, achieving superior performance across tested DE types and outperforming mainstream comparative methods.

## 2. Related Work

**Numerical methods based on ML** PINNs leverage the nonlinear representation power of neural networks to solve PDEs by embedding physical constraints directly into the training process as inductive biases (Raissi et al., 2019). This approach enables unsupervised learning relying on PDE residuals while faces several challenges. For example, their dependency on soft constraints for embedding physical laws may not thoroughly integrate exact physical priors and they often exhibit inefficiency when handling with stiff problems, characterized by rapidly changing solutions (Wang et al., 2021; Rao et al., 2023). To address these limitations, enhancements have been developed, including the hard-coding boundary conditions (Lu et al., 2021c), and employing temporal representations (Rao et al., 2023). In contrast, operator learning methods (Lu et al., 2021a; Cai et al., 2021; Li et al., 2023c;b) utilize neural networks to approximate nonlinear operators, providing a supervised learning approach to solving PDEs that depends on paired input-output data. While these methods offer significant potential, their substantial requirement for labeled data in scientific contexts poses a significant challenge. Both PINNs and neural operator learning methods yield approximate solutions to PDEs. These solutions frequently lack interpretability with respect to the underlying physical phenomena and demonstrate limited generalization beyond the specific scenarios used for training.

**Symbolic regression methods** Symbolic regression methods can be broadly categorized into two main categories: search-based and supervised learning-based approaches. Search-based methods explore the solution space to identify optimal symbolic expressions, encompassing traditional techniques such as genetic programming (GP) and heuristic search strategies (Forrest, 1993; Koza, 1994; Schmidt & Lipson, 2009; Staelens et al., 2013; Arnaldo et al., 2015; Iwo & Krawiec, 2019; Randall et al., 2022; Jiang & Xue, 2023). These conventional approaches, however, often grapple with an expansive search space, leading to computationally intensive tasks. Moreover, they tend to produce increasingly complex symbolic expressions without corresponding performance gains. To mitigate these challenges, recent advancements have integrated neural networks to guide and constrain the search process, enhancing efficiency (Sahoo et al., 2018; Zhang et al., 2023; Dong et al., 2024). Additionally, reinforcement learning techniques, including policy gradient methods and Monte Carlo tree searches, have been employed for symbolic regression (Petersen et al., 2020;

Mundhenk et al., 2021; Sun et al., 2023; Li et al., 2024). Although these methods exhibit slower inference times since they necessitate iterative searches during inference and the time-consuming evaluation of regressed symbolic expressions, they demonstrate promising performance in SRBench Black-box dataset (La Cava et al., 2021). In contrast, supervised learning-based approaches leverage large-scale pre-training on synthetic datasets to directly generate symbolic expressions (Biggio et al., 2021; Kamienny et al., 2022; Li et al., 2023a). By producing expressions in a single forward pass, these methods achieve significantly faster inference speeds compared to search-based techniques. A notable hybrid approach, TPSR (Shojaee et al., 2024), combines elements of search and supervised learning, yielding impressive performance. However, the effectiveness of supervised methods diminishes when the training data distribution diverges from that of the target physical system, leading to reduced model accuracy.

**ML for solving DEs analytically** Symbolic regression's ability to uncover expressive and interpretable formulations makes it a promising tool for deriving understandable symbolic solutions to DEs. Genetic algorithm-based symbolic regression methods have been employed to solve DEs (Tsoulos & Lagaris, 2006; Oh et al., 2023; Cao et al., 2023). Oh et al. introduce Bingo (Randall et al., 2022), using automatic differentiation to assess physical residuals in DEs as a fitness function to optimize symbolic expressions. Heuristic algorithms inspired by swarm intelligence, such as ant colony programming (Kamali et al., 2015) and artificial bee colony programming (Boudouaoui et al., 2020) emulate resource-seeking behaviors to search for solutions. However, these methods are computationally intensive, particularly for high-dimensional problems, where the expansive search space poses significant challenges. To address this, transfer learning has been explored to reduce the search space by leveraging solutions from single-dimensional instances of high-dimensional PDEs (HD-PDEs) (Cao et al., 2024). Yet, this approach is limited, as most physical systems governed by PDEs lack known single-dimensional equations. Liang and Yang (2022) propose a reinforcement learning pipeline to guide a weighted symbolic network toward approximate solution of DEs. The introduction of weights reduces interpretability and complicates the discovery of minimalist analytical expressions. Kolmogorov-Arnold Networks (KANs) (Liu et al., 2025) offer a novel approach for deriving symbolic solutions to DEs by incorporating a physics-regularized loss function. However, converting KANs' learned representations into symbolic expressions often degrades accuracy, resulting in solutions that fail to precisely satisfy the underlying equations. Majumdar et al. (2023) perform symbolic regression on numerical solutions from PINNs to identify interpretable expressions. Since PINN-derived solutions are inherently approximate,

the resulting symbolic expressions may not fully adhere to physical constraints, introducing inaccuracies due to error propagation.

## 3. Preliminary Studies

**Differential equations** Differential equations (DEs) are equations related to unknown univariate or multivariate functions and their derivatives or partial derivatives. Consider a bounded domain $\Omega \subset \mathbb{R}^n$, with a point $\mathbf{x} = (x_1, x_2, \ldots, x_n) \in \Omega$. Let $u(\mathbf{x})$ denote the unknown function to be determined. For a positive integer $k$, $D^k u \in \mathbb{R}^{n^k}$ represents all $k$-th order partial derivatives of $u$. Given a function $\mathcal{F} : \Omega \times \mathbb{R} \times \mathbb{R}^n \times \cdots \times \mathbb{R}^{n^{k-1}} \times \mathbb{R}^{n^k} \to \mathbb{R}$, the general form of a $k$-th order partial differential equation (PDE) can be expressed as in Eq. (1). When $n = 1$, the PDE reduces to an ordinary differential equation (ODE).

$$\mathcal{F}[\mathbf{x}, u(\mathbf{x}), Du(\mathbf{x}), \ldots, D^{k-1}u(\mathbf{x}), D^k u(\mathbf{x})] = 0 \quad (1)$$

Prominent examples include Poisson's equation:

$$-\Delta u(\mathbf{x}) = -\sum_{i=1}^{n} \frac{\partial^2 u}{\partial x_i^2} = f(\mathbf{x}), \quad (2)$$

defined on the spatial domain $\Omega$, and the heat equation:

$$\frac{\partial u(\mathbf{x}, t)}{\partial t} - a^2 \Delta u(\mathbf{x}, t) = f(\mathbf{x}, t), \quad a > 0, \quad (3)$$

defined on the spatiotemporal domain $(x, t) \in \Omega \times \tau$, where $a$ is a constant and $T > 0$. To ensure a well-posed problem for analytical solutions, a PDE must be accompanied by appropriate boundary conditions (BCs) and, for time-dependent PDEs, initial conditions (ICs). BCs are typically specified as $\mathcal{B}(u; \mathbf{x} \in \partial\Omega) = 0$ on the boundary $\partial\Omega$, while ICs for time-dependent problems are given as $\mathcal{I}(u; t = 0, \mathbf{x} \in \Omega) = 0$. Together, the differential equation, BCs, and ICs define a complete mathematical problem.

**Closed-form solution** The closed-form solution $\hat{u}(\mathbf{x})$ to a differential equation (DE) is composed of a finite combination of known functions, such as elementary functions (e.g., polynomials, exponentials, trigonometric functions), special functions (e.g., Bessel or hypergeometric functions), or, in some contexts, neural operators. Within a bounded domain, the solution must satisfy the DE along with its associated boundary conditions (BCs) and, for time-dependent problems, initial conditions (ICs). For example, $\hat{u} = c\cosh(x_1)\sin(x_2)$, involving the constant $c$, operators $\{\times, \cosh, \sin\}$ and variables $x_1$ and $x_2$, is a candidate closed-form solution to the Laplace equation $-\Delta u(\mathbf{x}) = 0$.

## 4. Methodology

SSDE identifies closed-form solutions using an unsupervised reinforcement learning framework, as depicted in Fig.

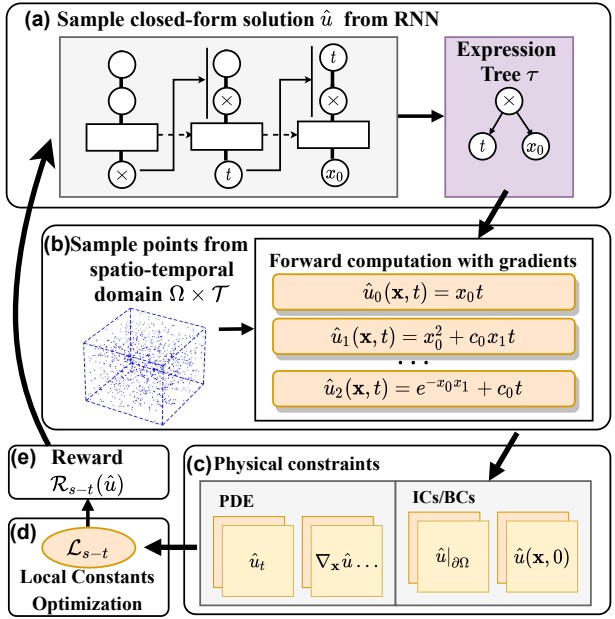

**(a) Sample closed-form solution $\hat{u}$ from RNN**

Expression Tree $\tau$

**(b) Sample points from spatio-temporal domain $\Omega \times \mathcal{T}$**

**Forward computation with gradients**

$$\hat{u}_0(\mathbf{x}, t) = x_0 t$$

$$\hat{u}_1(\mathbf{x}, t) = x_0^2 + c_0 x_1 t$$

$$\cdots$$

$$\hat{u}_2(\mathbf{x}, t) = e^{-x_0 x_1} + c_0 t$$

**(e) Reward**
$$\mathcal{R}_{s-t}(\hat{u})$$

**(c) Physical constraints**

PDE

$\hat{u}_t$  $\nabla_{\mathbf{x}}\hat{u} \cdots$

ICs/BCs

$\hat{u}|_{\partial\Omega}$  $\hat{u}(\mathbf{x}, 0)$

**(d)** $\mathcal{L}_{s-t}$
**Local Constants Optimization**

*Figure 1.* Algorithm overview (using spatiotemporal dynamical systems as an example). (a) The RNN generates skeletons for candidate solutions. (b) Sampled points are fed into the candidate skeletons to construct computational graphs with constants $c_i$ as parameters. (c) Physical constraints are built via automatic differentiation. (d) Constants are optimized to minimize $\mathcal{L}_{s-t}$. (e) The evaluator computes rewards based on physical constraints to train RNN. The process iterates until a valid solution is found.

1. To accelerate convergence while maintaining precision, we propose a risk-seeking constants optimization method. Additionally, for solving high-dimensional partial differential equations (HD-PDEs), we develop a recursive exploration technique. Pseudocode for the proposed algorithms is included in the appendix A.

### 4.1. Physics-Regularized Reinforcement Learning

**Expression skeleton generator**  We leverage symbolic expression trees, where internal nodes denote mathematical operators, and terminal nodes represent input variables or constants. The pre-order traversal of the corresponding expression tree $\tau$ represents the expression skeleton of $\hat{u}$. Each constant within the skeleton is a column vector of length $|c_{\text{dim}}|$. Therefore, when $|c_{\text{dim}}| \neq 1$, a single expression skeleton can yield multiple outputs for identical inputs, as detailed in Section 4.3. Inspired by automated machine learning (AutoML) works (Bello et al., 2017; Cuomo et al., 2022), we employ a Recurrent Neural Network (RNN) to generate categorical distributions over tokens from a given symbolic library $\mathcal{L}$. For each token $\tau_i$ (e.g., $+, \times, x_1, c, \dots$), we sample with likelihood $P(\tau_i|\tau_{1:(i-1)}; \theta)$ in an autoregressive manner to ensure contextual dependencies on previously chosen sym-

bols. The likelihood of the entire sampled expression $\hat{u}$ is computed as $p(\hat{u}|\theta) = \prod_{i=1}^{|\tau|} p(\tau_i|\tau_{1:(i-1)}; \theta)$, where $|\tau|$ is the length of the pre-order traversal. The RNN inputs include representations of the parent and sibling nodes of the token being sampled, reinforcing the model's understanding of the expression's tree structure.

**Physics-regularized local constant optimization(LCO)**  To ensure the expression skeleton complies with physical constraints, we employ PyTorch's automatic differentiation method (Paszke et al., 2017) to compute the differential term associated with the constraints. The constants in $\hat{u}$ are optimized by minimizing the discrepancy between predicted values and the given conditions. For instance, in a spatiotemporal dynamical system satisfying Dirichlet conditions, data points are sampled from the spatiotemporal domain $\Omega \times \mathcal{T}$. We use nonlinear optimization algorithms such as the BFGS algorithm (Fletcher, 2000), to optimize the constants by minimize the sum of mean squared errors $\mathcal{L}_{s-t}$ (Eq. (4), where 's-t' denotes the spatio-temporal system). The loss terms for the collocation points corresponding to the differential term $\mathcal{F}$ (defined in Eq. (1)), boundary conditions $\mathcal{B}$, and initial conditions $\mathcal{I}$, where $\{\mathbf{x}_f^i, t_f^i\}_{i=1}^{N_{\mathcal{F}}}$, $\{\mathbf{x}_b^i, t_b^i, u_b^i\}_{i=1}^{N_{\mathcal{B}}}$, $\{\mathbf{x}_0^i, u_0^i\}_{i=1}^{N_{\mathcal{I}}}$ represent the respective collocation points, and $\lambda_i$ denotes the relative weight of each loss term.

$$\begin{cases} \mathcal{L}_{s-t} = \lambda_0 \text{MSE}_{\mathcal{F}} + \lambda_1 \text{MSE}_{\mathcal{B}} + \lambda_2 \text{MSE}_{\mathcal{I}} \\ \text{MSE}_{\mathcal{F}} = \frac{1}{N_{\mathcal{F}}} \sum_i^{N_{\mathcal{F}}} |\mathcal{F}(x_f^i, t_f^i)|^2 \\ \text{MSE}_{\mathcal{B}} = \frac{1}{N_{\mathcal{B}}} \sum_i^{N_{\mathcal{B}}} |\hat{u}(x_b^i, t_b^i) - u_b^i|^2 \\ \text{MSE}_{\mathcal{I}} = \frac{1}{N_{\mathcal{I}}} \sum_i^{N_{\mathcal{I}}} |\hat{u}(x_0^i, 0) - u_0^i|^2 \end{cases} \quad (4)$$

**Solution Evaluator**  We conceptualize the task of discovering closed-form solutions as a Markov Decision Process (MDP) comprising the elements $(\mathcal{S}, \mathcal{A}, \mathcal{P}, \mathcal{R})$. The siblings and parent nodes of the current symbolic node act as the observation $\mathcal{S}$. The policy is defined by the distribution of tokens produced by the RNN, $p(\tau_i|\theta)$, from which the action (token $\tau_i$) is sampled, leading to a transition to a new expression state. Each generation of the expression is guided by terminal and undiscounted reward functions, where each episode reflects one cycle. To evaluate the generated expression under physics regularization, we apply a squashing function to the average of the root mean square errors (RMSE) of physical constraints across different systems. The reward for the expression skeleton $\hat{u}$ in a spatiotemporal dynamical system is computed as:

$$\mathcal{R}_{s-t}(\hat{u}) = \frac{1}{1 + \text{Average}(\sqrt{\text{MSE}_{\mathcal{F}}} + \sqrt{\text{MSE}_{\mathcal{B}}} + \sqrt{\text{MSE}_{\mathcal{I}}})} \quad (5)$$

**Training RNN using policy gradients**  Standard policy gradient methods typically focus on optimizing the average

performance of the policy, which deviates from the objective of finding an optimal closed-form solution. To maximize the best-case performance, we employ risk-seeking policy gradients with genetic programming introduced in (Petersen et al., 2020; Mundhenk et al., 2021), with the learning objective $J_{\text{risk}}(\theta; \epsilon)$ parameterized by $\epsilon$ as:

$$J_{\text{risk}}(\theta; \epsilon) \approx \mathbb{E}_{\hat{u} \sim p(\hat{u}|\theta)}[\mathcal{R}(\hat{u})|\mathcal{R}(\hat{u}) \geq \mathcal{R}_\epsilon(\theta)] \quad (6)$$

where $\mathcal{R}_\epsilon(\theta)$ is the reward distribution's $(1 - \epsilon)$-quantile under the current policy. To foster exploration, we integrate the hierarchical entropy of the sampled expressions into the reward term, weighted by $\lambda_\mathcal{H}$ (Landajuela et al., 2021):

$$\mathcal{H}(\theta) = \lambda_\mathcal{H} \mathbb{E}_{\hat{u} \sim p(\hat{u}|\theta)} \left[ \sum_{i=1}^{|\tau|} \gamma^{i-1} H[p(\tau_i | \tau_{1:(i-1)}; \theta)] \right] \quad (7)$$

### 4.2. Risk-Seeking Constant Optimization (RSCO)

Optimizing constants within the expression skeleton to satisfy physical constraints, particularly those related to differential equations (e.g., $\text{MSE}_\mathcal{F}$), incurs considerable computational costs. Building on the risk-seeking policy gradients introduced in (Petersen et al., 2020), the training objective focuses solely on the top $\epsilon$ fraction of episodes. Note that the skeleton must satisfy deterministic conditions to meet physical constraints. As $\hat{u}$ approaches these deterministic conditions, the likelihood of satisfying the differential equation's physical constraints increases. Therefore, we propose a risk-seeking constant optimization method.

In each iteration, the constants of a batch of sampled expression skeletons are optimized based on deterministic conditions by minimizing the modified loss $\mathcal{L}'_{\text{s-t}}$ (defined in Eq. (8)). The reward $\tilde{\mathcal{R}}$ for each expression is still computed by Eq. (5). The top $\epsilon$ fraction of samples incorporates all physical constraints as defined in Eq. (4), while the constants are refined and the rewards are recalculated to yield more accurate evaluations. The accurate rewards of the precise top $\epsilon$ fraction are then used to calculate the policy gradient $\nabla_\theta J_{\text{risk}}(\theta, \epsilon)$ via Eq. (9), which will guide the RNN. By enforcing boundary or initial conditions before optimization, this strategy eliminates unnecessary constant optimization based on differential relationships for expression skeletons that already violate physical constraints, echoing the philosophy of hard constraint embedding. (Lu et al., 2021c).

$$\mathcal{L}'_{\text{s-t}} = \lambda_1 \text{MSE}_\mathcal{B} + \lambda_2 \text{MSE}_\mathcal{I} \quad (8)$$

$$\nabla_\theta J_{\text{risk}}(\theta, \epsilon) = \mathbb{E}_{\hat{u} \sim p(\hat{u}|\theta)}[(\mathcal{R}(\hat{u}) - \mathcal{R}_\epsilon(\theta)) \cdot$$
$$\nabla_\theta \log p(\hat{u}|\theta)|\tilde{\mathcal{R}}(\hat{u}) \geq \tilde{\mathcal{R}}_\epsilon(\theta)] \quad (9)$$

### 4.3. Recursion-Based Exploration

As the solution dimensionality increases, the symbol library expands, leading to an exponential growth in the search space. The pre-order traversal length $|\tau|$, and the size of the symbol library $|\mathcal{L}|$ result in a combinatorial search space of size $|\mathcal{L}|^{|\tau|}$. The computational complexity of solving HD-PDEs also grows with dimensionality, due to the increased number of variables in the physical constraints. To mitigate this, we propose a recursive exploration scheme that decomposes the solution of multidimensional differential equations into sequential, dependent rounds, with each round focusing on a single variable.

**Parametric expression** We focus on finding the expression skeleton for the single variable $\{x_i\}_{i=1}^d$. The terms formulated using other variables $\mathbf{x}_{-i}$ are treated as fluctuating parameters $\vec{\alpha}(\mathbf{x}_{-i}) \in \mathbb{R}^m$, where $m$ denotes the number of such parameters. The skeleton $\hat{u}(\mathbf{x})$ is transformed into a parametric expression $\tilde{u}(x_i; \vec{\alpha}(\mathbf{x}_{-i}))$. For the first $k$ observed variable, we can get the solution a parametric form: $\hat{u} = \tilde{u}(x_1, x_2, \cdots, x_k; \vec{\alpha}_k(\mathbf{x}_{-\{1,2,\ldots,k\}}))$. Here, $\mathbf{x}_{-\{1,2,\ldots,k\}}$ represents all variables except $x_1, x_2, \ldots, x_k$.

**Recursive call** When $k = 0$, the close-form solution is viewed as a parameter $\hat{u} = \vec{\alpha}_0 = \alpha_0$. For each round $k > 0$, we update the expression for the $k$-th observed variable $x_k$. The last round's parametric expressions $\vec{\alpha}_{k-1}$ are viewed as new parametric expressions $\tilde{u}_k$ dependent on $x_k$ and the new parameters $\vec{\alpha}_k(\mathbf{x}_{-\{\ldots,k\}})$. Namely, $\vec{\alpha}_{k-1} = \tilde{u}_k(x_k; \vec{\alpha}_k(\mathbf{x}_{-\{\ldots,k\}}))$, which forms a typical recursive relation.

**Recursive case** We ascertain the parametric skeletons $\tilde{u}_k$ on the single dimension, as well as the parameter vector $\vec{\alpha}_k$ by the reinforcement learning method proposed above. In the parametric expression, each parameter is a vector of length $|c_{\text{dim}}|$, corresponding to the varying values calculated from data points collected within the $\mathbf{x}_{-i}$ space. Here, $|c_{\text{dim}}|$ represents the number of points. These parameters are optimized in accordance with the specified physical constraints, and the resulting parametric expressions are evaluated. If the variance of the $|c_{\text{dim}}|$ dimensional parameters falls below a predetermined threshold, we surmise that this parameter does not vary with $\mathbf{x}_{-i}$ and it degenerates to a constant. This approach allows us to distinguish between fixed constants and $\vec{\alpha}$ parameters.

**Base case** When the identified parametric skeleton has no remaining parameters, $\tilde{u}$ is considered a symbol expression independent of the other variables. In the final round, the parameter expression from the previous round, $\vec{\alpha}_{d-1}$, becomes a symbolic expression dependent only on the last variable $x_d$. Further, the closed-form solution to a $d$-dim DE is given as:

$$\hat{u} = \tilde{u}(x_1, x_2, \cdots, x_k; \vec{\alpha}_k(\mathbf{x}_{-\{1,2,\ldots,k\}}))$$
$$= \tilde{u}_1(x_1; \tilde{u}_2(x_2; \ldots \tilde{u}_d(x_d; \vec{c})))$$

Ultimately, by substituting all parameters with their corresponding symbolic expressions regarding the other variables and further optimizing the constants within the expression skeleton based on physical constraints, we can derive a closed-form solution for the multidimensional differential equation.

The recursive process enables SSDE to learn in a focused manner, addressing one variable dimension at a time.

# 5. Experimental Settings

We present experimental settings used to evaluate SSDE by answering the following research questions (**RQs**):

**RQ1:** How does SSDE compare to other mainstream methods in finding closed-form solutions to differential equations?

**RQ2:** Can RSCO and the recursive exploration strategies improve SSDE's ability to find closed-form solutions?

**RQ3:** Does SSDE produce more accurate solutions than performing symbolic regression methods based on numerical solutions?

All details regarding datasets, computing infrastructures, baselines and their parameter settings, along with additional comparative experiments and results, are reported in appendix.

## 5.1. Metrics

The evaluation of our methodology and baselines is conducted from two distinct perspectives:

- The mean Root Mean Squared Error (RMSE) between the predicted closed-form solution and physical constraints, denoted by $\mathcal{L}_{\text{PHY}}$. Taking the spatiotemporal dynamical system as an example and referring to Eq. (4), it is defined as:

$$\mathcal{L}_{\text{PHY}} = \text{Average}(\sqrt{\text{MSE}_{\mathcal{F}}} + \sqrt{\text{MSE}_{\mathcal{B}}} + \sqrt{\text{MSE}_{\mathcal{I}}}) \tag{10}$$

  This measure assesses whether the identified solutions are consistent with physical constraints.

- The complete recovery rate of closed-form solution expressions. For a given closed-form solution $u$, if the skeleton of the found expression $\hat{u}$ after simplification is exactly the same or equivalent to $u$, and after the final optimization of constants, the mean square error between the expression and the true solution is less than $1 \times 10^{-8}$, the expression is considered to have completely recovered the true solution. $P_{\text{RE}}$ represents the proportion of closed-form solution expressions that

completely recover the true solution in 20 independent experiments. This metric quantifies the symbolic and numerical differences between the regressed expression and the true solution.

## 5.2. Baselines

We evaluate the performance of the proposed recursive exploration method by comparing it with the following methods used for seeking the closed-form solutions:

- **PINN+DSR**: Performs symbolic regression with DSR (Petersen et al., 2020) on the numerical solutions obtained from PINNs (Raissi et al., 2019).

- **Kolmogorov–Arnold Networks (KAN)**: A recently proposed method designed to replace Multi-Layer Perceptrons (MLP) (Liu et al., 2025). Compared to MLP, KAN performs better in symbolic formula representation tasks (Yu et al., 2024) and shows promise for discovering closed-form solutions to HD-PDEs.

- **PR-GPSR**: Uses genetic programming to search for a function that satisfies the given PDE and boundary conditions (Oh et al., 2023).

## 5.3. Benchmark Problem Sets

To evaluate the performance of our method across a diverse range of differential equations, we selected the following equations for testing:

- **2D and 3D Poisson's Equations**: As a classic example of time-independent partial differential equation, Poisson's equation is widely used in fields such as electrostatics, fluid dynamics, and mechanical engineering.

- **2D and 3D Heat Equations**: These equations are canonical examples of spatio-temporal dynamic systems, describing the phenomena related to heat transfer.

- **2D and 3D Nonlinear Wave Equations**: Nonlinear wave equations are fundamental models for describing the propagation of waves in various media, where the wave amplitude depends on both space and time.

The equations and their deterministic conditions are provided in Table 1.

# 6. Results on Benchmarks

**(RQ1) Solution accuracy on benchmarks** Table 2 presents a comparative analysis of solutions from SSDE against other methods used to derive closed-form solutions. SSDE consistently outperforms the other methods across all benchmarks, discovering solutions with significantly smaller

*Table 1.* Summary of tested differential equations with their deterministic conditions

| NAME | PDE | DETERMINISTIC CONDITIONS |
|---|---|---|
| POISSON2D | $\frac{\partial^2 u}{\partial x_1^2}+\frac{\partial^2 u}{\partial x_2^2}=30x_1^2-7.8x_1+1,\mathbf{X}\in[-1,1]^2$ | $u(\mathbf{X})=2.5x_1^4-1.3x_1^3+0.5x_2^2-1.7x_2,\mathbf{X}\in\partial[-1,1]^2$ |
| POISSON3D | $\frac{\partial^2 u}{\partial x_1^2}+\frac{\partial^2 u}{\partial x_2^2}+\frac{\partial^2 u}{\partial x_3^2}=30x_1^2-7.8x_2+1,\mathbf{X}\in[-1,1]^3$ | $u(\mathbf{X})=2.5x_1^4-1.3x_2^3+0.5x_3^2,\mathbf{X}\in\partial[-1,1]^3$ |
| HEAT2D | $\frac{\partial u}{\partial t}-(\frac{\partial^2 u}{\partial x_1^2}+\frac{\partial^2 u}{\partial x_2^2})=-30x_1^2+7.8x_2+t,\mathbf{X}\in[-1,1]^2$ | $u(\mathbf{X},t)=2.5x_1^4-1.3x_2^3+0.5t^2,\mathbf{X}\in\partial[-1,1]^2,t\in[0,1]$ 
 $u(\mathbf{X},0)=2.5x_1^4-1.3x_2^3,\mathbf{X}\in[-1,1]^2$ |
| HEAT3D | $\frac{\partial u}{\partial t}-(\frac{\partial^2 u}{\partial x_1^2}+\frac{\partial^2 u}{\partial x_2^2}+\frac{\partial^2 u}{\partial x_3^2})=-30x_1^2+7.8x_2-2.7$ 
 $,\mathbf{X}\in[-1,1]^3$ | $u(\mathbf{X},t)=2.5x_1^4-1.3x_2^3+0.5x_3^2-1.7t,\mathbf{X}\in\partial[-1,1]^3,t\in[0,1]$ 
 $u(\mathbf{X},0)=2.5x_1^4-1.3x_2^3+0.5x_3^2,\mathbf{X}\in[-1,1]^3$ |
| WAVE2D | $\frac{\partial^2 u}{\partial t^2}-(\frac{\partial^2 u}{\partial x_1^2}+\frac{\partial^2 u}{\partial x_2^2})=-u-u^3+((0.25-4x_1^2)\cdot$ 
 $e^{x_1^2-0.5t}+e^{3x_1^2-1.5t}\sin(x_2)^2)\sin(x_2),\mathbf{X}\in[-1,1]^2$ | $u(\mathbf{X},t)=\exp(x_1^2)\sin(x_2)e^{-0.5t},\mathbf{X}\in\partial[-1,1]^2,t\in[0,1]$ 
 $u(\mathbf{X},0)=\exp(x_1^2)\sin(x_2),\mathbf{X}\in[-1,1]^2$ |
| WAVE3D | $\frac{\partial^2 u}{\partial t^2}-(\frac{\partial^2 u}{\partial x_1^2}+\frac{\partial^2 u}{\partial x_2^2}+\frac{\partial^2 u}{\partial x_3^2})=u^2-((4x_1^2+4x_3^2+2.75)\cdot$ 
 $e^{x_1^2+x_3^2-0.5t}+e^{2x_1^2+2x_3^2-t}\cos(x_2))\cos(x_2),\mathbf{X}\in[-1,1]^3$ | $u(\mathbf{X},t)=\exp(x_1^2+x_3^2)\cos(x_2)e^{-0.5t},\mathbf{X}\in\partial[-1,1]^3,t\in[0,1]$ 
 $u(\mathbf{X},0)=\exp(x_1^2+x_3^2)\cos(x_2),\mathbf{X}\in[-1,1]^3$ |

*Table 2.* Comparison of average $\mathcal{L}_{\text{PHY}}(\hat{u})$ and recovery rates among SSDE and baselines for solving differential equations. 95% confidence intervals are obtained from the standard error between mean $\mathcal{L}_{\text{PHY}}(\hat{u})$ on each problem set.

| | SSDE | | PINN+DSR | | KAN | | PR-GPSR | |
|---|---|---|---|---|---|---|---|---|
| NAME | $\mathcal{L}_{\text{PHY}}\downarrow$ | $P_{\text{RE}}\uparrow$ | $\mathcal{L}_{\text{PHY}}\downarrow$ | $P_{\text{RE}}\uparrow$ | $\mathcal{L}_{\text{PHY}}\downarrow$ | $P_{\text{RE}}\uparrow$ | $\mathcal{L}_{\text{PHY}}\downarrow$ | $P_{\text{RE}}\uparrow$ |
| POISSON2D | **7.20E-5±2.57E-6** | **100%** | 5.71E-1±7.88E-2 | 0% | 7.20E+0±1.46E+0 | 0% | 1.90E-2±2.32E-3 | 0% |
| POISSON3D | **5.70E-5±1.33E-6** | **100%** | 2.60E+0±1.11E-1 | 0% | 6.16E+0±1.52E+0 | 0% | 1.80E-1±1.76E-2 | 0% |
| HEAT2D | **1.47E-6±7.64E-7** | **100%** | 3.21E+0±1.35E-1 | 0% | 1.16E+1±1.24E+0 | 0% | 2.15E-1±1.28E-2 | 0% |
| HEAT3D | **1.05E-5±1.23E-6** | **100%** | 2.69E+0±3.24E-1 | 0% | 1.15E+1±1.72E+0 | 0% | 8.87E-1±2.96E-1 | 0% |
| WAVE2D | **4.25E-5±2.35E-6** | **100%** | 9.56E-1±2.14E-1 | 0% | 2.04E+0±1.42E-1 | 0% | 4.24E-1±3.69E-1 | 0% |
| WAVE3D | **8.17E-6±1.56E-6** | **100%** | 1.34E+1±4.16E+0 | 0% | 1.36E+1±2.38E+0 | 0% | 9.87E-1±2.46E-1 | 0% |

navigate the exponentially expanded search space to find accurate closed-form solutions. In contrast, SSDE's recursive exploration strategy, which employs reinforcement learning independently for each dimension, exhibits a remarkable advantage in solving higher-dimensional PDEs. Specifically, SSDE reliably identifies correct closed-form solutions for both linear and nonlinear differential equations. We also provide a detailed illustration of how SSDE progressively isolates each variable and recursively constructs the correct solution through stepwise regression in the appendix F.

**(RQ2) Ablation studies** We conducted a series of ablation studies to evaluate the individual contributions of key components, including RSCO and the recursive exploration strategy (RecurExp). Figure 2 illustrates the complete recovery rate of SSDE on the PDE benchmarks under the same execution time for each ablated configuration. The results highlight the critical role of these components in enhancing overall performance. Specifically, the absence of RSCO significantly impairs SSDE's ability to identify correct closed-form solutions within the same time frame. Similarly, disabling the recursive exploration strategy drastically diminishes SSDE's capability to solve high-dimensional partial differential equations, preventing it from discovering accurate closed-form solutions.

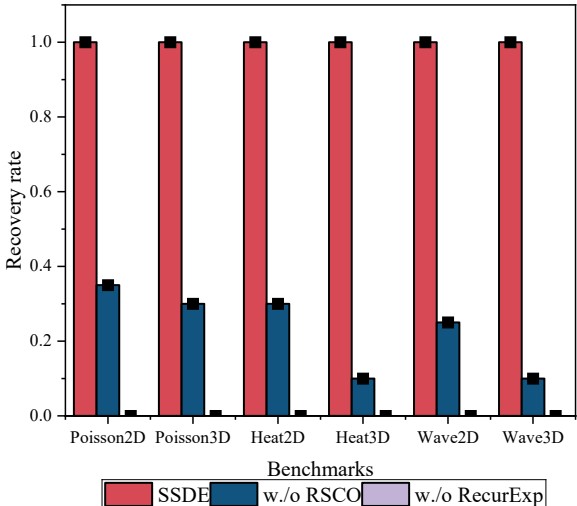

*Figure 2.* Ablation study of SSDE on benchmarks of PDE.

$L_{\text{PHY}}$ values. The increased dimensionality poses a substantial challenge for the other methods, as they struggle to

*Table 3.* Identified Solutions $\hat{u}$ by SSDE on DEs

| NAME | GROUND TRUTH | IDENTIFIED SOLUTION $\hat{u}$ |
|------|--------------|------------------------------|
| POISSON2D | $x_1^2(2.5x_1^2-1.3x_1)+0.5x_2(x_2-1.4)-x_2$ | $x_1^2(2.5000x_1^2-1.3000x_1)+0.4999x_2(x_2-1.4002)-x_2$ |
| POISSON3D | $2.5x_1^4-1.3x_2^3+0.5x_3^2$ | $2.5000x_1^4-1.3000x_2^3+0.4999x_3^2+2.3763e{-}5$ |
| HEAT2D | $0.5t^2+2.5x_1^4-1.3x_2^3$ | $0.5000t^2+\log(e^{2.5000x_1^4-x_2^2(1.3000x_2-8.1811e-7)})$ |
| HEAT3D | $-1.7t+2.5x_1^4-1.3x_2^3+0.5x_3^3$ | $-1.7003t+2.5000x_1^4-1.3000x_2^3+0.4999x_3^3+0.0002$ |
| WAVE2D | $\exp(x_1^2)\sin(x_2)e^{-0.5t}$ | $\exp(-0.5000t+x_1^2)\sin(x_2)$ |
| WAVE3D | $\exp(x_1^2+x_3^2)\cos(x_2)e^{-0.5t}$ | $\exp(-0.5000t+1.000x_1^2+x_3^2)\cos(x_2)^{1.000}$ |

*Table 4.* Comparison of average $\mathcal{L}_{\text{PHY}}(\hat{u})$ and recovery rates between SSDE and PINN+DSR for Poisson's equations within $\Gamma$ dataset. 95% confidence intervals are obtained from the standard error between mean $\mathcal{L}_{\text{PHY}}(\hat{u})$ on each problem set.

| NAME | DIFFERENTIAL EQUATION | SSDE | | PINN+DSR | |
|------|----------------------|------|------|----------|------|
| | | $\mathcal{L}_{\text{PHY}}\downarrow$ | $P_{\text{RE}}\uparrow$ | $\mathcal{L}_{\text{PHY}}\downarrow$ | $P_{\text{RE}}\uparrow$ |
| $\Gamma_1$ | $\frac{\partial^2 u}{\partial x_1^2}=6x_1+2$ | **8.70E-8±7.72E-9** | 100% | 2.93E-5±2.36E-6 | 100% |
| $\Gamma_2$ | $\frac{\partial^2 u}{\partial x_1^2}=12x_1^2+6x_1+2$ | **3.65E-7±2.62E-7** | 100% | 6.55E-5±5.78E-6 | 100% |
| $\Gamma_3$ | $\frac{\partial^2 u}{\partial x_1^2}=20x_1^3+12x_1^2+6x_1+2$ | **3.98E-7±9.22E-8** | 100% | 2.52E-4±3.38E-5 | 100% |
| $\Gamma_4$ | $\frac{\partial^2 u}{\partial x_1^2}=30x_1^4+20x_1^3+12x_1^2+6x_1+2$ | **7.24E-7±7.79E-8** | **100%** | 2.47E-3±7.83E-4 | 0% |
| $\Gamma_5$ | $\frac{\partial^2 u}{\partial x_1^2}=(2\cos(x_1)-4x_1\sin(x_1))\cos(x_1^2)-(4x_1^2+1)\cdot$ $\sin(x_1^2)\cos(x_1)$ | **1.39E-6±2.06E-6** | 100% | 8.66E-5±4.05E-6 | 100% |
| $\Gamma_6$ | $\frac{\partial^2 u}{\partial x_1^2}=2\cos(x_1^2+x_1)-\sin(x_1)-(2x_1+1)^2\sin(x_1^2+x_1)$ | **7.60E-7±8.75E-7** | 100% | 1.26E-5±8.48E-7 | 100% |
| $\Gamma_7$ | $\frac{\partial^2 u}{\partial x_1^2}=-4x_1^2/(x_1^2+1)^2+2/(x_1^2+1)-1/(x_1+1)^2$ | **4.06E-6±5.82E-6** | **100%** | 8.42E-3±1.15E-3 | 0% |
| $\Gamma_8$ | $\frac{\partial^2 u}{\partial x_1^2}=-0.25/x_1^{1.5}$ | **1.23E-6±2.01E-6** | **100%** | 1.98E-5±1.87E-6 | 70% |
| $\Gamma_9$ | $\frac{\partial^2 u}{\partial x_1^2}+\frac{\partial^2 u}{\partial x_2^2}=2\cos(x_2^2)-4x_2^2\sin(x_2^2)-\sin(x_1)$ | **3.22E-7±3.27E-8** | **100%** | 5.15E-3±1.32E-3 | 0% |
| $\Gamma_{10}$ | $\frac{\partial^2 u}{\partial x_1^2}+\frac{\partial^2 u}{\partial x_2^2}=-4\sin(x_1)\cos(x_2)$ | **1.27E-6±1.92E-6** | **100%** | 2.58E-3±3.17E-4 | 0% |
| $\Gamma_{11}$ | $\frac{\partial^2 u}{\partial x_1^2}+\frac{\partial^2 u}{\partial x_2^2}=x_1^{x_2}\log(x_1)^2+x_1^{x_2}x_2(x_2-1)/x_1^2$ | **1.24E-5±2.29E-5** | **100%** | 4.89E-3±2.69E-3 | 0% |
| $\Gamma_{12}$ | $\frac{\partial^2 u}{\partial x_1^2}+\frac{\partial^2 u}{\partial x_2^2}=12x_1^2-6x_1+1.0$ | **1.22E-5±2.32E-5** | **100%** | 6.68E-1±1.48E-1 | 0% |

## 7. (RQ3) Compared with SR on Numerical Solutions

To further demonstrate the superiority of SSDE over symbolic regression methods that operate on numerical solutions, we conduct a series of benchmarks based on the Nguyen dataset, adapted for Poisson's equations with known source terms. Each differential equation is designed so that its analytical solution corresponds to one of the expressions in the Nguyen dataset (Uy et al., 2011). The baseline method first computes numerical solutions for these equations using PINNs and then applies DSR directly to the numerical outputs.

Table 4 reports the results of SSDE and the baseline method in terms of $\mathcal{L}_{\text{PHY}}$ and recovery rate $P_{\text{RE}}$. Due to the approximation errors inherent in numerical solutions and the compounding of these errors during the symbolic regression process, the baseline methods occasionally recover the correct solution skeletons, but frequently yield significant inaccuracies during constant optimization. In contrast,

SSDE bypasses the reliance on numerical surrogates entirely and consistently recovers accurate closed-form expressions across all test cases.

## 8. Conclusion

We propose SSDE, a reinforcement learning-based framework for discovering closed-form symbolic solutions to differential equations. SSDE employs a recurrent neural network to generate symbolic candidates, guided by an evaluator grounded in physical constraints. To improve learning efficiency and convergence, we introduce a risk-seeking constant optimization technique. By formulating symbolic solutions as parametric expressions, SSDE can decompose high-dimensional PDEs into recursively solvable, single-dimensional components. This recursive formulation allows SSDE to efficiently recover closed-form solutions to complex differential equations. Extensive experiments on various types of differential equations demonstrate that SSDE can discover accurate symbolic solutions without

prior mathematical knowledge, offering a promising new direction for analytical DE solving.

## Software and Data

To facilitate reproducibility and further research, we provide the complete implementation of SSDE, including training scripts, benchmark configurations, and symbolic expression evaluation tools. The source code is publicly available at:

https://github.com/Hintonein/SSDE

In addition, all benchmark datasets constructed for this study, including the high-dimensional PDEs and Poisson problems derived from Nguyen expressions, are included in the repository. The code is implemented in Python and relies on PyTorch and standard scientific computing libraries.

## Acknowledgement

This work was supported by National Natural Science Foundation of China (No.92370117) and the CAS Project for Young Scientists in Basic Research (No.YSBR-090).

## Impact Statement

This paper presents work whose goal is to advance the field of Machine Learning. There are many potential societal consequences of our work, none which we feel must be specifically highlighted here.

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

# A. Pseudocode for SSDE

In this section, we present the pseudocode of SSDE. The overall procedure is described in Alg. 1, and the recursive exploration mechanism is shown in Alg. 3. Additionally, we provide the subroutines `SampleSkeleton` (line 5 in Alg. 1), `ReplaceParameters` (line 12 in Alg. 3), and `SingleDimensionalSSDE` (line 9 in Alg. 3) in Algs. 2, 4, and 5, respectively.

Alg. 1 describes the overall SSDE framework with the risk-seeking constant optimization (RSCO) strategy. An RNN generates candidate symbolic skeletons, and the constants within each skeleton are optimized using boundary and initial conditions. A reward is then computed based on physical constraints. After collecting a batch of rewards, the $(1-\epsilon)$-quantile is computed to select top-performing skeletons. These selected expressions are further refined by re-optimizing constants, and the updated rewards are used to compute risk-seeking policy gradients and entropy regularization for RNN parameter updates. This process continues until the termination criterion is met. Alg. 2 defines the symbolic skeleton sampling process using the RNN. The `ApplyConstraints` function (line 8) filters infeasible tokens to reduce the search space, while `Arity` (line 11) returns the argument count for each token. `ParentSibling` (line 12) calculates the parent and sibling nodes to guide the next token generation. This top-down sampling process follows a similar mechanism as DSR (Petersen et al., 2020); readers are referred to the original paper for further details.

In Alg. 3, we propose a recursive algorithm based on single-dimensional SSDE (described in Alg. 5) to identify the closed-form solutions to high-dimensional partial differential equations (HD-PDEs). For an HD-PDE with $d$ variables, the algorithm iteratively represents the symbolic solution as a hierarchical parametric traversal. At iteration $k$, the traversal corresponding to the $k$-th dimension is denoted as $\tilde{\tau}^{(k)}(x_k; \vec{\alpha}_k)$ about single variable $x_k$ and containing parameters $\vec{\alpha}_k$, where $\vec{\alpha}_k$ contains parameters dependent on subsequent variables $\mathbf{x}_{-\{\cdots,k\}}$. Initially, the entire solution is regarded as a

---

**Algorithm 1** SSDE: Reinforcement Learning Framework for Discovering Closed-Form PDE Solutions with RSCO

**Input**: differential equation $\mathcal{F}$; deterministic conditions (boundary conditions $\mathcal{B}$, initial conditions $\mathcal{I}$)
**Parameter**: RNN learning rate $\eta$; entropy coefficient $\lambda_{\mathcal{H}}$; risk factor $\epsilon$; batch size $N$; sample epochs $M$; reward function $R$
**Output**: identified closed-form solution $\hat{u}^*$ that best satisfies the physical constraints

1: Initialize RNN with parameters $\theta$
2: Randomly sample the domain to build the dataset $\mathcal{D} = \{\mathbf{x}_f^i\}_{i=1}^{N_{\mathcal{F}}} \cup \{\mathbf{x}_b^i, u_b^i\}_{i=1}^{N_{\mathcal{B}}} \cup \{\mathbf{x}_0^i, u_0^i\}_{i=1}^{N_{\mathcal{I}}}$
3: **for** $i \leftarrow 1$ **to** $M$ **do**
4:     **for** $j \leftarrow 1$ **to** $N$ **do**
5:         $\hat{u}_j \leftarrow \text{SampleSkeleton}(\theta)$         ▷ Sample a skeleton
6:         $(\mathcal{L}'_{\text{s-t}})_j \leftarrow \text{BIConstraintsCalc}(\hat{u}_j, \mathcal{D})$         ▷ Calculate the physics-regularized loss concerning $\mathcal{B}, \mathcal{I}$
7:         $\hat{u}'_j \leftarrow \text{BFGS}(\hat{u}_j, \mathcal{D}, (\mathcal{L}'_{\text{s-t}})_j)$         ▷ Optimize constants in the skeleton using BFGS
8:         $\tilde{R}(\hat{u}'_j) \leftarrow \text{RewardCalc}(\hat{u}'_j)$         ▷ Compute the reward
9:     **end for**
10:     $\tilde{\mathcal{R}} \leftarrow \{\tilde{R}(\hat{u}'_j)\}_{j=1}^N$         ▷ Collect batch rewards
11:     $\tilde{R}_\epsilon \leftarrow (1-\epsilon)$-quantile of $\tilde{\mathcal{R}}$         ▷ Compute reward threshold
12:     $\mathcal{U} : \{\hat{u}_q\} \leftarrow \{\hat{u}'_j : \tilde{R}(\hat{u}'_j) \geq \tilde{R}_\epsilon\}$         ▷ Select expressions with rewards exceeding the $\epsilon$-quantile threshold
13:     **for** $q \leftarrow 1$ **to** $\epsilon \cdot N$ **do**
14:         $(\mathcal{L}_{\text{s-t}})_q \leftarrow \text{PrLoss}(\hat{u}_q, \mathcal{D})$         ▷ Calculate the physics-regularized loss
15:         $\hat{u}'_q \leftarrow \text{BFGS}(\hat{u}_q, \mathcal{D}, (\mathcal{L}_{\text{s-t}})_q)$         ▷ Optimize constants in the skeleton using BFGS
16:         $R(\hat{u}'_q) \leftarrow \text{RewardCalc}(\hat{u}'_q)$         ▷ Compute precise reward
17:     **end for**
18:     $\mathcal{R} \leftarrow \{R(\hat{u}'_q)\}_{q=1}^{\epsilon * N}$         ▷ Collect precise batch rewards
19:     $\hat{g}_1 \leftarrow \text{ReduceMean}((\mathcal{R} - R_\epsilon)\nabla_\theta \log p(\mathcal{U}|\theta))$         ▷ Compute risk-seeking policy gradient
20:     $\hat{g}_2 \leftarrow \text{ReduceMean}(\lambda_{\mathcal{H}} \nabla_\theta \mathcal{H}(\mathcal{U}|\theta))$         ▷ Compute entropy gradient
21:     $\theta \leftarrow \theta + \eta(\hat{g}_1 + \hat{g}_2)$         ▷ Apply gradients
22:     **if** $\max \mathcal{R} > R(\hat{u}^*)$ **then**
23:         $\hat{u}^* \leftarrow \hat{u}^{\arg\max \mathcal{R}}$         ▷ Update best expression
24:     **end if**
25: **end for**

---

**Algorithm 2** Sampling a closed-form solution's skeleton from the RNN

---

**Input**: RNN with parameters $\theta$; tokens library $\mathcal{L}$

**Output**: The closed-form solution's skeleton $\hat{u}$ from the RNN

1:  $\tau \leftarrow []$                                           $\triangleright$ Initialize empty pre-order traversal of skeleton

2:  counter $\leftarrow 1$                              $\triangleright$ Initialize counter for number of unselected nodes

3:  $x \leftarrow$ empty$||$empty                     $\triangleright$ Initial RNN input is empty parent and sibling

4:  $C_0 \leftarrow \vec{0}$                                    $\triangleright$ Initialize RNN cell state to zero

5:  $i \leftarrow 1$

6:  **while** counter $\neq 0$ **do**

7:     $(\psi_i, c_i) \leftarrow$ RNN$(x, C_{i-1}; \theta)$                  $\triangleright$ Emit probabilities; update state

8:     $\psi_i \leftarrow$ ApplyConstraints$(\psi_i, \mathcal{L}, \tau)$              $\triangleright$ Adjust probabilities

9:     $\tau_i \leftarrow$ Categorical$(\psi_i)$                   $\triangleright$ Sample next token

10:    $\tau \leftarrow \tau || \tau_i$                            $\triangleright$ Append token to traversal

11:    counter $\leftarrow$ counter $+$ Arity$(\tau_i) - 1$        $\triangleright$ Update number of unselected nodes

12:    $x \leftarrow$ ParentSibling$(\tau)$               $\triangleright$ Compute next parent and sibling

13:    $i \leftarrow i + 1$

14:  **end while**

15:  $\hat{u} \leftarrow \tau$             $\triangleright$ Assemble the pre-order traversal $\tau$ into an expression skeleton

16:  **return** $\hat{u}$

---

**Algorithm 3** Recursive Exploration based on single-dimensional SSDE

---

**Input**: Differential equation $\mathcal{F}$; deterministic conditions (boundary conditions $\mathcal{B}$, initial conditions $\mathcal{I}$).

**Parameter**: Number of variables in differential equations $d$.

**Output**: Identified closed-form solution $\hat{u}^*$ that best satisfies the physical constraints.

1:  Randomly sample the domain to build the dataset $\mathcal{D} = \{\mathbf{x}_f^i\}_{i=1}^{N_{\mathcal{F}}} \cup \{\mathbf{x}_b^i, u_b^i\}_{i=1}^{N_{\mathcal{B}}} \cup \{\mathbf{x}_0^i, u_0^i\}_{i=1}^{N_{\mathcal{I}}}$

2:  Treat the solution as a parameter $\alpha_0$, initialize $\alpha_0$ with $\mathcal{B}, \mathcal{I}$

3:  $\hat{\tau}^{(0)}(x_0; \alpha_0) \leftarrow \alpha_0$            $\triangleright$ Initialize empty regressed expression's traversal with $\alpha_0$

4:  $|\vec{\alpha}_0| \leftarrow 1$                     $\triangleright$ Initialize number of parameters of $\hat{\tau}^{(0)}$ as 1

5:  $\hat{\tau} \leftarrow \hat{\tau}^{(0)}(x_0; \alpha_0)$         $\triangleright$ Initialize pre-order traversal of regressed skeleton with $\hat{\tau}^{(0)}$

6:  **for** $k \leftarrow 1$ **to** $d$ **do**

7:     $\tilde{\tau}^{(k)}(x_k; \vec{\alpha}_k) \leftarrow []$

8:     **for** $m \leftarrow 1$ **to** $|\vec{\alpha}_{k-1}|$ **do**

9:        $\tilde{\tau}_k^{(m)}(x_k; \vec{\alpha}_k^m) \leftarrow$ SingleDimSSDE$(\mathcal{D}, \vec{\alpha}_{k-1}[m], k)$    $\triangleright$ Seek parametric expressions for each $\alpha_{d-1}$

10:       $\tilde{\tau}_k(x_k; \vec{\alpha}_k) \leftarrow \tilde{\tau}_k(x_k; \vec{\alpha}_k) || \tilde{\tau}_k^{(m)}(x_k; \vec{\alpha}_k^m)$    $\triangleright$ Append new parametric expression of $k$th variable

11:     **end for**

12:    $\tau \leftarrow$ RepalceParameters$(\hat{\tau}(\mathbf{x}_{0:k-1}; \vec{\alpha}_{k-1}), \tilde{\tau}_k)$  $\triangleright$ Replace parameters in $\tau$ with new sub-expressions discovered for the $k$th variable

13:    **if** $k \neq d$ **then**

14:       $\mathcal{L}_{\text{s-t}}^{(k)} \leftarrow$ PartialConstraintsCalc$(\tau(\mathbf{x}_{0:k}; \vec{\alpha}_k), \mathcal{D})$    $\triangleright$ Calculate constraints across the regressed dimensions

15:       $\hat{\tau} \leftarrow$ BFGS$(\tau, \mathcal{D}, \mathcal{L}_{\text{s-t}}^{(k)})$    $\triangleright$ Optimize parameters and constants in the skeleton with BFGS

16:    **else**

17:       $\mathcal{L}_{\text{s-t}} \leftarrow$ PhysicalConstraintsCalc$(\tau(\mathbf{x}_{0:d}), \mathcal{D})$    $\triangleright$ Calculate the physics-regularized loss

18:       $\tau^* \leftarrow$ BFGS$(\tau, \mathcal{D}, \mathcal{L}_{\text{s-t}})$    $\triangleright$ Optimize constants in the skeleton with BFGS

19:    **end if**

20:  **end for**

21:  $\hat{u}^* \leftarrow \tau^*$           $\triangleright$ Assemble the pre-order traversal $\tau$ into an expression skeleton

22:  **return** $\hat{u}^*$

---

single parameter $\alpha_0$, dependent on all variables $\mathbf{x}_{1:d}$. We formally denote this as a trivial traversal $\hat{\tau}^{(0)}(x_0, \alpha_0)$, where $x_0$ is is merely a placeholder without practical meaning, simplifying to $\hat{\tau}^{(0)} = \alpha_0$. At each subsequent iteration $k$, Alg. 5 generates single-dimensional traversals $\tilde{\tau}_k(x_k; \vec{\alpha}_k)$ for parameters $\vec{\alpha}_{k-1}$ identified in the previous step. Alg. 4 then substitutes the previous parameters $\vec{\alpha}_{k-1}$ within the current traversal $\hat{\tau}$ with these newly obtained expressions, producing the updated

---

**Algorithm 4** Replace parameters in regressed skeleton with $k$th variable's traversals

---

**Input**: Regressed skeleton $\hat{\tau}(\mathbf{x}_{0:k-1}; \vec{\alpha}_{k-1})$; new identified traversals $\tilde{\tau}_k$ of $k$-th variable .
**Output**: New traversal $\tau$ of symbolic skeleton of $k$ variables.

1: $\tau = []$ ▷ Initialize empty traversal of $k$ variables
2: **for** $i \leftarrow 1$ **to** $|\hat{\tau}(\mathbf{x}_{0:k-1}; \vec{\alpha}_{k-1})|$ **do**
3:     **for** $m \leftarrow 1$ **to** $|\vec{\alpha}_{k-1}|$ **do**
4:         **if** $\hat{\tau}[i] = \vec{\alpha}^m_{k-1}$ **then**
5:             $\tau \leftarrow \tau || \tilde{\tau}_k[m]$ ▷ Replace parameter $\vec{\alpha}^m_{k-1}$ with the corresponding traversal of the $k$th variable
6:         **else**
7:             $\tau \leftarrow \tau || \hat{\tau}[i]$ ▷ Append regressed traversal
8:         **end if**
9:     **end for**
10: **end for**
11: **return** $\tau$

---

traversal $\tau$ involving $d$ variables. To mitigate error propagation during the recursion, constants in the traversal $\tau$ are refined by minimizing $\mathcal{L}^{(k)}_{\text{s-t}}$, which incorporates partial physical constraints. At the final iteration ($k = d$), when the traversal involves the last variable $x_d$ without further parameters, we optimize the traversal $\tau$ against the complete physical constraints $\mathcal{L}_{\text{s-t}}$, thereby obtaining the final precise symbolic expression.

The underlying principle of Alg. 5 closely parallels that of Alg. 1, yet specifically targets discovering symbolic traversals corresponding to single-dimensional variable skeletons. Here, we employ an RNN to generate candidate traversals $[\tilde{\tau}_k]_j$ associated with the $k$-th variable during each training batch. Each constant token (c) within these traversals is represented as a vector $[\vec{p}_k]_j$ of dimensionality $|c\text{dim}|$. Initially, the vectors are optimized by minimizing deterministic constraints sampled from boundary ($\mathcal{B}$) and initial conditions ($\mathcal{I}$).

We denote the $h$-th c token within traversal $[\tilde{\tau}_k]_j$ as $[\vec{p}_k]^h_j$, thus $|c\text{dim}| = |[\vec{p}_k]^h_j|$. If the variance of a given vector $[\vec{p}_k]^h_j$ falls below a predefined threshold $\delta$, it is treated as a constant; otherwise, it remains as a parameter. The parameters retained in $[\vec{p}_k]_j$ subsequently become new parameters for traversal $[\tilde{\tau}_k]_j$. As in Alg. 1, we utilize the RSCO strategy to select the top-$\epsilon$ traversals within the batch, computing rewards accordingly. Finally, the RNN parameters are updated using risk-seeking policy gradients augmented by entropy regularization.

As an illustrative example, consider discovering the closed-form solution:

$$u = 2.5x_1^4 - 1.3x_2^3 + 0.5x_3^2$$

Initially, we represent the symbolic solution simply as $\hat{u} = \alpha_0$. The expression skeleton corresponding to the first variable $x_1$ is identified using Alg. 5, yielding a parametric traversal:

$$\tilde{u}(x_1; \vec{\alpha}_1) = c_1 x_1^4 + \alpha_1$$

where the parameter $\alpha_1$ depends on subsequent variables $x_2$ and $x_3$. Next, Alg. 5 identifies the skeleton for the second variable $x_2$:

$$\alpha_1(x_2; \vec{\alpha}_2) = c_1 x_2^3 + \alpha_2$$

Using Alg. 4, we combine the skeletons derived from the first two dimensions to form the intermediate traversal:

$$\hat{u} = c_1 x_1^4 + c_2 x_2^3 + \alpha_2$$

where $\alpha_2$ remains dependent on $x_3$. Subsequently, the traversal for the final variable $x_3$ is obtained as

$$\alpha_2(x_3) = c_1 x_3^2$$

Combining all traversals results in the complete symbolic expression

$$\hat{u} = c_1 x_1^4 + c_2 x_1^3 + c_3 x_2^2$$

Finally, optimizing the constants $c_1, c_2, c_3$ against the full set of physical constraints precisely recovers the exact closed-form solution.

---

**Algorithm 5** Single-dimensional SSDE

---

**Input**: Differential equation $\mathcal{F}$; deterministic conditions (boundary conditions $\mathcal{B}$, initial conditions $\mathcal{I}$); the $m$th last regressed variable's paramter $\vec{\alpha}_{d-1}[m]$;the order $d$th of the variable currently in focus

**Parameter**: RNN learning rate $\eta$; entropy coefficient $\lambda_{\mathcal{H}}$; risk factor $\epsilon$; batch size $N$; sample epochs $M$; reward function $R$; early stop threshold $R_t$; constant judgement threshold $\delta$

**Output**: Pre-order traversal $\tilde{\tau}_k^*$ of the parametric expression of $k$-th variable that best satisfies the physical constraints

1: Randomly sample the domain to build the dataset $\mathcal{D} = \{\mathbf{x}_f^i\}_{i=1}^{N_{\mathcal{F}}} \cup \{\mathbf{x}_b^i, u_b^i\}_{i=1}^{N_{\mathcal{B}}} \cup \{\mathbf{x}_0^i, u_0^i\}_{i=1}^{N_{\mathcal{I}}}$
2: Initialize generator RNN with parameters $\theta$
3: **for** $i \leftarrow 1$ **to** $M$ **do**
4:     **for** $j \leftarrow 1$ **to** $N$ **do**
5:         $[\tilde{\tau}_k]_j \leftarrow \text{SampleSkeleton}(\theta)$                              ▷ Sample $k$-th variable's expression skeleton
6:         $[\tilde{\mathcal{L}}'_{\text{s-t}}]_j^k \leftarrow \text{SingleDimBIConstraints}([\tilde{\tau}_k]_j, \mathcal{D}, \vec{\alpha}_{k-1}[m])$      ▷ Calculate single-dimensional constraints on $\mathcal{B}, \mathcal{I}$
7:         $[\tilde{\tau}_k]'_j(x_k, [\vec{p}_k]_j) \leftarrow \text{BFGS}([\tilde{\tau}_k]_j, \mathcal{D}, \vec{\alpha}_{k-1}[m], [\tilde{\mathcal{L}}'_{\text{s-t}}]_j^k)$    ▷ Optimize parameters in the skeleton using BFGS
8:         **for** $h \leftarrow 1$ **to** $|[\vec{p}_k]_j|$ **do**
9:             **if** $\text{variance}([\vec{p}_k]_j^h) < \delta$ **then**
10:                 $[\vec{p}_k]_j^h \leftarrow \text{average}([\vec{p}_k]_j^h)$            ▷ Fix parameter to average value if its variance falls below threshold $\delta$
11:             **end if**
12:         **end for**
13:         $\vec{\alpha}_k \leftarrow \{[\vec{p}_k]_j | [\vec{p}_k]_j^h \text{ is not const.}\}$             ▷ Extract variable parameters from optimized constants
14:         $\tilde{R}([\tilde{\tau}_k]'_j) \leftarrow \text{RewardCalc}([\tilde{\tau}_k]'_j)$                     ▷ Compute the reward
15:     **end for**
16:     $\tilde{\mathcal{R}} \leftarrow \{\tilde{R}([\tilde{\tau}_k]'_j)\}_{j=1}^N$                            ▷ Compute batch rewards
17:     $\tilde{R}_\epsilon \leftarrow (1-\epsilon)\text{-quantile of } \tilde{\mathcal{R}}$                   ▷ Compute reward threshold
18:     $\mathcal{T} : \{[\tilde{\tau}_k]_q\} \leftarrow \{[\tilde{\tau}_k]'_j : \tilde{R}([\tilde{\tau}_k]'_j) \geq \tilde{R}_\epsilon\}$    ▷ Select expressions with rewards exceeding the $\epsilon$-quantile threshold
19:     **for** $q \leftarrow 1$ **to** $\epsilon \cdot N$ **do**
20:         $[\tilde{\mathcal{L}}_{\text{s-t}}]_q^k \leftarrow \text{SingleDimPhysicalConstraints}([\tilde{\tau}_k]_q, \mathcal{D}, \vec{\alpha}_{k-1}[m])$      ▷ Calculate the physics-regularized loss
21:         $[\tilde{\tau}_k]'_q(x_k, [\vec{p}_k]_q) \leftarrow \text{BFGS}([\tilde{\tau}_k]_q, \mathcal{D}, \vec{\alpha}_{k-1}[m], [\tilde{\mathcal{L}}_{\text{s-t}}]_q^k)$    ▷ Optimize constants in the skeleton using BFGS
22:         **for** $h \leftarrow 1$ **to** $|[\vec{p}_k]_q|$ **do**
23:             **if** $\text{variance}([\vec{p}_k]_q^h) < \delta$ **then**
24:                 $[\vec{p}_k]_q^h \leftarrow \text{average}([\vec{p}_k]_q^h)$            ▷ Fix parameter to average value if its variance falls below threshold $\delta$
25:             **end if**
26:         **end for**
27:         $\vec{\alpha}_k \leftarrow \{[\vec{p}_k]_q | [\vec{p}_k]_q \text{ is not const.}\}$             ▷ Extract variable parameters from optimized constants
28:         $R([\tilde{\tau}_k]'_q) \leftarrow \text{RewardCalc}([\tilde{\tau}_k]'_q)$                    ▷ Compute the reward
29:     **end for**
30:     $\mathcal{R} \leftarrow \{R([\tilde{\tau}_k]'_q)\}_{k=1}^{\epsilon * N}$                         ▷ Compute precise batch rewards
31:     $\hat{g}_1 \leftarrow \text{ReduceMean}((\mathcal{R} - R_\epsilon)\nabla_\theta \log p(\mathcal{T}|\theta))$        ▷ Compute risk-seeking policy gradient
32:     $\hat{g}_2 \leftarrow \text{ReduceMean}(\lambda_{\mathcal{H}} \nabla_\theta \mathcal{H}(\mathcal{U}|\theta))$               ▷ Compute entropy gradient
33:     $\theta \leftarrow \theta + \eta(\hat{g}_1 + \hat{g}_2)$                                  ▷ Apply gradients
34:     **if** $\max \mathcal{R} > R([\tilde{\tau}_k]^*)$ **then**
35:         $[\tilde{\tau}_k]^* \leftarrow [\tilde{\tau}_k]^{\arg \max \mathcal{R}}$                     ▷ Update best expression
36:     **end if**
37: **end for**

---

## B. Computing Infrastructure and Hyperparameter Settings

In this section, we provide additional experimental details, including the computing infrastructure and the specific hyperparameter configurations for SSDE.

**Computing infrastructure** All experiments reported in this work were conducted on an Intel(R) Xeon(R) Gold 6138 CPU @ 2.00GHz. The algorithm implementation can also leverage GPU acceleration for improved computational efficiency.

*Table 5.* Hyperparameters for SSDE.

| HYPERPARAMETER | SYMBOL | VALUE |
|---|---|---|
| LEARNING RATE | $\eta$ | 0.0010 |
| ENTROPY WEIGHT | $\lambda_{\mathcal{H}}$ | 0.07 |
| ENTROPY GAMMA | $\gamma$ | 0.7 |
| RNN CELL SIZE | – | 32 |
| RNN CELL LAYERS | – | 1 |
| RISK FACTOR | $\epsilon$ | 0.05 |
| MAX EPOCH | $M$ | 200 |
| BATCH SIZE | $N$ | 1000 |
| PDE CONSTRAINT WEIGHT | $\lambda_0$ | 1 |
| BC CONSTRAINT WEIGHT | $\lambda_1$ | 1 |
| IC CONSTRAINT WEIGHT | $\lambda_2$ | 1 |

*Table 6.* Convergent relative $\mathcal{L}_2$ error of baselines on high-dimensional dataset

| NAME | SSDE | MATHEMATICA | PR-GPSR | KAN | FEX | DSR | PINN |
|---|---|---|---|---|---|---|---|
| POISSON2D | **1.55E-05** | ✗ | 1.49E-02 | 3.24E+00 | 2.00E-02 | 9.85E-02 | 6.24E-04 |
| POISSON3D | **3.58E-06** | ✗ | 2.01E-02 | 2.57E+00 | 9.13E-02 | 2.15E-01 | 3.93E-03 |
| HEAT2D | **4.45E-06** | ✗ | 4.57E-02 | 6.54E+00 | 4.70E-02 | 2.19E-01 | 6.98E-03 |
| HEAT3D | **9.34E-06** | ✗ | 1.39E+00 | 5.58E+00 | 3.63E-01 | 3.20E-01 | 1.15E-02 |
| WAVE2D | **3.11E-05** | ✗ | 1.80E-01 | 1.18E+00 | 2.34E-01 | 7.40E-02 | 1.34E-02 |
| WAVE3D | **7.62E-06** | ✗ | 4.56E-01 | 7.26E+00 | 3.97E-01 | 1.95E+00 | 3.69E-02 |

**Hyperparameter settings** The hyperparameters of our method mainly include the hyperparameters for the reinforcement learning policy gradient optimization, and weighting coefficients of the physical constraints defined in the loss function $\mathcal{L}_{\text{s-t}}$:

$$\mathcal{L}_{\text{s-t}} = \lambda_0 \text{MSE}_{\mathcal{G}} + \lambda_1 \text{MSE}_{\mathcal{B}} + \lambda_2 \text{MSE}_{\mathcal{I}} \tag{11}$$

For the policy optimizer, we considered the following hyperparameter search spaces: learning rate $\in$ $\{0.0001, 0.0005, 0.0010\}$, entropy weight $\lambda_{\mathcal{H}} \in \{0.04, 0.07, 0.10\}$, and hierarchical entropy's coefficient $\gamma \in \{0.5, 0.7, 0.9\}$. Hyperparameter tuning was performed via grid search on benchmark problems $\Gamma_5$ and $\Gamma_{10}$ from the $\Gamma$ dataset, and the best hyperparameters were selected by minimizing the average $\mathcal{L}_{PHY}$. The optimal hyperparameters identified through this tuning process are summarized in Table 5, and these configurations were consistently used across all benchmark experiments.

## C. Additional Experiments and Results

### C.1. Experiments with Additional Baselines

To facilitate fair comparison across different methods, we adopt the relative $\mathcal{L}_2$ error (defined by Eq. (12)) as a unified evaluation metric in our supplementary experiments.

$$\text{Mean Relative } L_2 \text{ Loss} = \frac{1}{n} \sum_{i=1}^{n} \frac{\|\mathbf{y}_{\text{true},i} - \mathbf{y}_{\text{pred},i}\|_2}{\|\mathbf{y}_{\text{true},i}\|_2} \tag{12}$$

In the main text, we compare SSDE with the baseline method **PINN+DSR**, which performs symbolic regression on numerical solutions of differential equations. Additionally, we evaluate both methods on a suite of PDE benchmarks. To further enrich the comparison, we introduce two additional baselines. The first is **Mathematica** (Wolfram Research, Inc., 2024), a commercial software capable of computing analytical solutions to differential equations. The second is **FEX** (Liang & Yang, 2022), a reinforcement learning-based framework for solving differential equations. Detailed descriptions of these two baselines are provided in Appendix D. Table 6 summarizes the performance of all methods across the PDE benchmarks. SSDE consistently identifies correct closed-form solutions and outperforms all baseline methods in terms of accuracy.

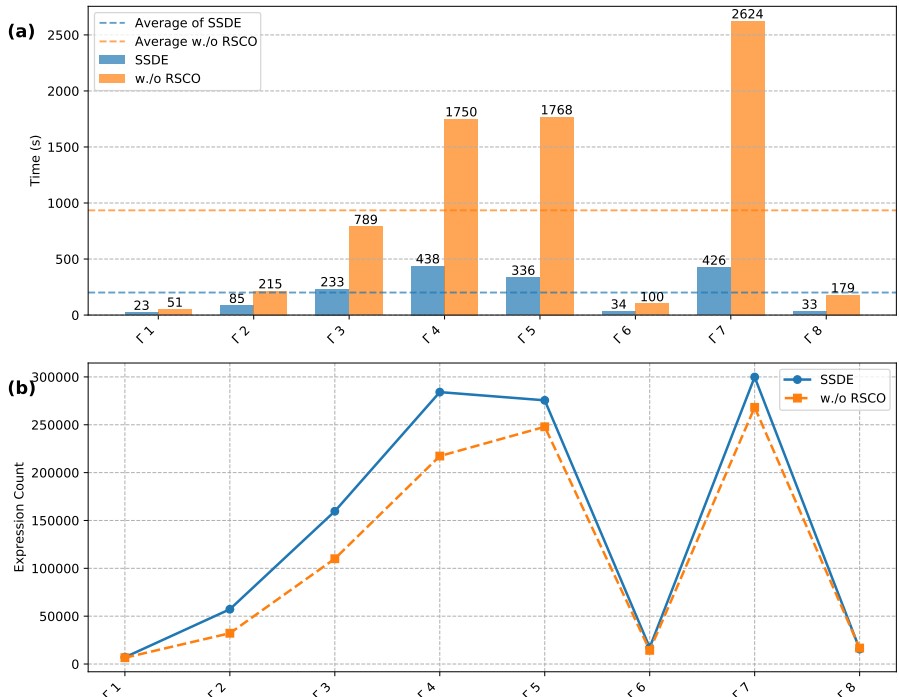

*Figure 3.* Ablation study on RSCO efficiency (results averaged over 20 benchmark runs) (a) Time efficiency comparison: RSCO achieves 4.65× speedup ($\mu = 934$s vs. 200s) in solution discovery time. (b) Search space comparison: While requiring 22% more expression evaluations, RSCO maintains superior time efficiency.

### C.2. Complexity Analysis

We conduct a complexity analysis to address two research questions (RQs):

**RQ1** Does the proposed risk-seeking constant optimization (RSCO) algorithm accelerate the discovery of closed-form solutions?

**RQ2** How does the size of the symbolic library affect the computational complexity of the algorithm?

To answer RQ1, we perform 20 independent trials on each of the $\Gamma_1$-$\Gamma_8$ benchmarks, comparing the average convergence time and number of iterations required with and without RSCO. As shown in Figure 3, RSCO consistently reduces the total solving time across all benchmarks, despite only marginally improving the average number of iterations. Interestingly, under the same random seed, RSCO sometimes achieves convergence in fewer iterations, highlighting the benefit of prioritizing boundary condition satisfaction early in the optimization process.

To investigate RQ2, we evaluate the impact of symbolic library size on solving complexity using the $\Gamma_1$–$\Gamma_4$ benchmarks. As shown in Figure 4, we start with a minimal operator set $+, -, \times, \div$ and progressively expand it by adding transcendental functions such as $\sin$, $\cos$, $\exp$, and $\log$. We observe that enlarging the symbolic library significantly increases the average time and number of iterations required to recover the correct solution. Moreover, the addition of higher-order operators like $\exp$ introduces a much larger increase in complexity than trigonometric functions. This is likely due to the presence of high-order polynomial terms in the target solutions, which can be easily confused with exponential or logarithmic expressions during symbolic search, especially given the close derivative relationships among $e^x$, $\ln x$, and certain polynomial approximations.

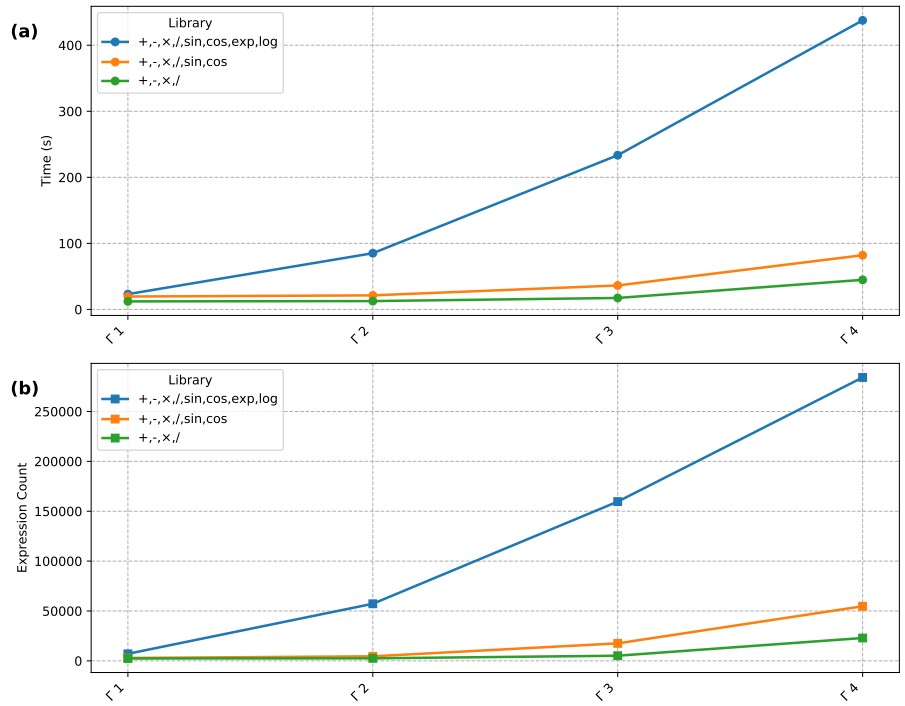

*Figure 4.* Computational complexity scaling with library size. Average (a) time and (b) explored expression count for SSDE in $\Gamma$1-4 experiments across 20 independent runs.

## D. Details of Baselines Algorithms

In this section, we introduce the mainstream methods used for identifying the closed-form symbolic solution to high-dimensional PDEs.

**Kolmogorov–Arnold Networks (KAN)**    A newly proposed method designed to replace Multi-Layer Perceptrons(MLP) incorporates capabilities for automatic symbolic regression(Liu et al., 2025). Compared with MLP, it performs better on symbolic formula representation tasks(Yu et al., 2024) and is promising for discovering closed-form solutions to HD-PDEs. Moreover, compared to EQL((Sahoo et al., 2018)), which is a fully-connected work where elementary functions are used as activation functions, this method does not require additional sparsification for symbolic regression and only requires a small number of nodes to complete the task of fitting the data. In this work, we use the open-source code and implementation provided by the authors[1]. The open source code has been implemented for the 2D Poisson's equation merely. We have also implemented the algorithm for the 3D Poisson's equation, 2D and 3D heat equations based on the open source code. It is imperative to note that due to the automatic symbolic regression process, expressions involving basic operations such as addition, subtraction, multiplication, and division can be directly derived from KAN's network structure. Therefore, the symbol library used in the KAN method for automatic regression does not include these operators. Additionally, `cos` token can also be directly obtained from `sin` by adding a bias (i.e., $\sin(x + \frac{\pi}{2}) = \cos(x)$), therefore, the `cos` token is also absent from the symbol library. To ensure a fair comparison, apart from the symbol library employed in our algorithm, we also introduced $\{\circ^2, \circ^3, \circ^4\}$ as additional symbols for use in the KAN method. In addition, we utilized the same hyperparameters and network architecture settings as those employed in the KAN repository's source code on partial differential equations.

**Physics-regularized Genetic Programming Based Symbolic Regression(PR-GPSR)**    PR-GPSR((Oh et al., 2023)) is a physics-regularized genetic programming(GP) based symbolic regression method for analytical solutions to differential equations. They use the mean-squared error with physical constraints as the fitness of population to measure how well

---

[1] https://github.com/KindXiaoming/pykan

the found expression $\hat{u}$ satisfies the differential equations. In this work, we use the open-source code and implementation provided by the authors[2]. The open source code has been implemented for the 2D Poisson's equation. We have also implemented the PR-GPSR algorithm for the 3D Poisson's equation, 2D and 3D heat equations based on the open source code. To ensure fair comparison, we use the hyperparameters settings reported in the original paper. In addition, we use the same symbol library as our algorithm and add the `CONST` token to give it the ability to generate constants.

**PINN+DSR**  To demonstrate that SSDE has better solution accuracy than the natural paradigm of directly applying symbolic regression algorithms to numerical solutions, we designed this baseline. It consists of two parts: the PINN algorithm and the DSR algorithm. The PINN algorithm is used to generate numerical solutions for each equation in the Γ dataset, and the DSR algorithm directly regresses the corresponding symbolic expressions based on the numerical solutions. The following will introduce these two algorithms and their implementation in detail.

**Physics-Informed Neural Network (PINN)**  A neural network method proposed by (Raissi et al., 2019), which is widely used to solve the partial differential equations numerically. PINN directly uses a fully connected neural network to approximate the solution function of the partial differential equation. Its numerical solution has high accuracy when solving low-dimensional PDEs with uncomplicated computational domains and boundary conditions. In this work, we use the open-source code and implementation by DeepXDE[3](Lu et al., 2021b). To ensure fair and effective comparison, we train the network with PDEs and deterministic conditions defined in Γ dataset until the network convergent. And sample the points in the computational domain identically to our approach used for subsequent symbolic regression. The convergent mean relative $\mathcal{L}_2$ error on the Γ dataset is shown in Table 7.

**Deep Symbolic Regression (DSR)**  DSR((Petersen et al., 2020)) is a symbolic regression method based on reinforcement learning. The risk-seeking policy gradient is used to guide the RNN to generate the symbolic expressions corresponding to the given observational data. In this work, we use the open-source code and implementation provided by the authors[4]. To ensure fair comparison, we use the same library as our algorithm and add the `CONST` token to give it the ability to generate constants. In addition, we used the hyperparameter combination reported in table 5 for the policy optimizer.

**Finite Expression Method (FEX)**  FEX is a reinforcement learning-based method for solving high-dimensional partial differential equations by discovering solutions expressed as finite mathematical expressions (Liang & Yang, 2022). Instead of parameterizing the solution with neural networks, FEX formulates the PDE-solving process as a combinatorial optimization problem over the space of mathematical expressions, represented as binary trees with both discrete operators and continuous parameters. While FEX is theoretically designed to recover compact, interpretable expressions that approximate the true solution, in practice, its symbolic structures are often coarse approximations, and the accuracy is largely achieved by tuning the continuous parameters. As a result, the discovered expressions tend to resemble parameterized symbolic networks, wherein most of the fitting power comes from weight optimization rather than symbolic structure. This compromises interpretability and makes it difficult to recover exact closed-form solutions, especially when the symbolic search fails to capture the core structure of the ground truth. Nonetheless, FEX still presents a promising approach for symbolic approximation of PDEs with lower memory cost and offers partial interpretability compared to black-box neural solvers. In this work, we use the official open-source implementation of FEX provided by the authors[5]. The repository includes support for 2D and 3D Poisson equations. To enable fair and comprehensive comparison, we further extend the implementation to support 2D and 3D heat equations and wave equations.

**Mathematica**  Mathematica is a commercial symbolic computation software developed by Wolfram Research, widely used for solving mathematical problems involving algebraic manipulation, symbolic integration, and differential equations. It provides a built-in solver for obtaining analytical solutions to a wide range of ordinary and partial differential equations, leveraging rule-based rewriting systems and symbolic inference engines. In contrast to data-driven approaches, Mathematica relies on curated mathematical knowledge and a comprehensive symbolic engine to derive closed-form solutions. However, Mathematica's performance is strongly tied to the internal heuristics and rule libraries it employs, which are generally optimized for classical PDEs with known solution templates. For high-dimensional PDEs, equations with nonstandard

---

[2]`https://github.com/HongsupOH/physics-regularized-bingo`
[3]`https://github.com/lululxvi/deepxde`
[4]`https://github.com/dso-org/deep-symbolic-optimization`
[5]`https://github.com/LeungSamWai/Finite-expression-method`

*Table 7.* Convergent mean relative $\mathcal{L}_2$ error of PINN on $\Gamma$ dataset

| NAME | EPOCH | MEAN RELATIVE $\mathcal{L}_2$ ERROR |
|---|---|---|
| $\Gamma_1$ | 30000 | 2.25E-05 |
| $\Gamma_2$ | 30000 | 6.32E-06 |
| $\Gamma_3$ | 30000 | 7.04E-05 |
| $\Gamma_4$ | 30000 | 5.28E-05 |
| $\Gamma_5$ | 30000 | 2.57E-05 |
| $\Gamma_6$ | 30000 | 2.14E-05 |
| $\Gamma_7$ | 30000 | 1.08E-05 |
| $\Gamma_8$ | 30000 | 1.05E-05 |
| $\Gamma_9$ | 50000 | 2.59E-03 |
| $\Gamma_{10}$ | 50000 | 5.91E-04 |
| $\Gamma_{11}$ | 50000 | 3.61E-04 |
| $\Gamma_{12}$ | 50000 | 2.88E-03 |

*Table 8.* Configuration of high-dimensional PDEs dataset.

| NAME | DOMAIN | LIBRARY $\mathcal{L}^*$ | GROUND TRUTH |
|---|---|---|---|
| POISSON2D | $[-1,1]^2$ | $\mathcal{L}_0 \cup \{x_2\}$ | $u(\mathbf{X}) = 2.5x_1^4 - 1.3x_1^3 + 0.5x_2^2 - 1.7x_2$ |
| POISSON3D | $[-1,1]^3$ | $\mathcal{L}_0 \cup \{x_2, x_3\}$ | $u(\mathbf{X}) = 2.5x_1^4 - 1.3x_2^3 + 0.5x_3^2$ |
| HEAT2D | $[0,1] \times [-1,1]^2$ | $\mathcal{L}_0 \cup \{x_2, t\}$ | $u(\mathbf{X},t) = 2.5x_1^4 - 1.3x_2^3 + 0.5t^2$ |
| HEAT3D | $[0,1] \times [-1,1]^3$ | $\mathcal{L}_0 \cup \{x_2, x_3, t\}$ | $u(\mathbf{X},t) = 2.5x_1^4 - 1.3x_2^3 + 0.5x_3^2 - 1.7t$ |
| WAVE2D | $[0,1] \times [-1,1]^2$ | $\mathcal{L}_0 \cup \{x_2, t\}$ | $u(\mathbf{X},t) = \exp(x_1^2)\sin(x_2)e^{-0.5t}$ |
| WAVE3D | $[0,1] \times [-1,1]^3$ | $\mathcal{L}_0 \cup \{x_2, x_3, t\}$ | $u(\mathbf{X},t) = \exp(x_1^2 + x_3^2)\cos(x_2)e^{-0.5t}$ |

$^*$ *Owing to the structure of the KAN network, the symbol library employed in the KAN baseline encompasses variables and constants, as well as the operators $\{\circ^2, \circ^3, \circ^4, \sin, \log, \exp\}$.*

boundary conditions, or synthetic benchmarks designed outside its built-in library scope, Mathematica often fails to return exact symbolic solutions or may produce no result at all. In this work, we use Mathematica (Version 14.2) as a baseline to assess its symbolic solving capabilities on our PDE benchmarks.

## E. Details of Datasets

### E.1. Dataset on High-dimensional PDEs

Despite the availability of existing datasets for validating numerical PDE solvers based on deep learning methods (Takamoto et al., 2022), these datasets typically do not provide explicitly defined symbolic solutions. Consequently, evaluating symbolic-solution discovery algorithms on such datasets is not only computationally expensive and inefficient but also does not guarantee the correctness or interpretability of the resulting symbolic expressions.

To comprehensively assess the effectiveness of our algorithm in discovering diverse symbolic solutions, we constructed a new benchmark dataset consisting of various types of differential equations (DEs). This dataset enables rigorous performance evaluation of our proposed method and other state-of-the-art symbolic-solution discovery algorithms. Specifically, each benchmark example (see Table 8) includes the differential equation itself, the computational domain, associated boundary conditions, and a symbol library defining permissible functional forms. For clarity and brevity, we define symbol libraries relative to a base library $\mathcal{L}_0$, given by $\mathcal{L}_0 = \{+, -, \times, \div, \sin, \cos, \exp, \log, x_1, \text{const.}\}$. The notation $[-1,1]^d$ denotes a spatial domain represented by a $d$-dimensional hypercube, while the notation $[0,1] \times [-1,1]^d$ explicitly differentiates between the temporal domain (the first interval) and the spatial domain (the second interval). The benchmark differential equations encompass a range of scenarios, including both stationary and spatiotemporal dynamics, as well as linear and nonlinear PDEs, thereby providing comprehensive coverage for validating symbolic solution discovery methods.

*Table 9.* Configuration of $\Gamma$ dataset on Poisson's Equation

| NAME | DOMAIN | LIBRARY $\mathcal{L}$ | GROUND TRUTH |
|------|--------|------------|--------------|
| $\Gamma_1$ | $[-1, 1]$ | $\mathcal{L}_0$ | $\mathbf{u}(\mathbf{x}) = x_1^3 + x_1^2 + x_1$ |
| $\Gamma_2$ | $[-1, 1]$ | $\mathcal{L}_0$ | $\mathbf{u}(\mathbf{x}) = x_1^4 + x_1^3 + x_1^2 + x_1$ |
| $\Gamma_3$ | $[-1, 1]$ | $\mathcal{L}_0$ | $\mathbf{u}(\mathbf{x}) = x_1^5 + x_1^4 + x_1^3 + x_1^2 + x_1$ |
| $\Gamma_4$ | $[-1, 1]$ | $\mathcal{L}_0$ | $\mathbf{u}(\mathbf{x}) = x_1^6 + x_1^5 + x_1^4 + x_1^3 + x_1^2 + x_1$ |
| $\Gamma_5$ | $[-1, 1]$ | $\mathcal{L}_0$ | $\mathbf{u}(\mathbf{x}) = \sin(x_1^2)\cos(x_1) - 1$ |
| $\Gamma_6$ | $[-1, 1]$ | $\mathcal{L}_0$ | $\mathbf{u}(\mathbf{x}) = \sin(x_1) + \sin(x_1 + x_1^2)$ |
| $\Gamma_7$ | $[0.5, 1.5]$ | $\mathcal{L}_0$ | $\mathbf{u}(\mathbf{x}) = \log(x_1 + 1) + \log(x_1^2 + 1)$ |
| $\Gamma_8$ | $[0.5, 1.5]$ | $\mathcal{L}_0$ | $\mathbf{u}(\mathbf{x}) = \sqrt{x_1}$ |
| $\Gamma_9$ | $[0.5, 1.5]^2$ | $\mathcal{L}_0 \cup \{x_2\}$ | $\mathbf{u}(\mathbf{x}) = \sin(x_1) + \sin(x_2^2)$ |
| $\Gamma_{10}$ | $[0.5, 1.5]^2$ | $\mathcal{L}_0 \cup \{x_2\}$ | $\mathbf{u}(\mathbf{x}) = 2\sin(x_1)\cos(x_2)$ |
| $\Gamma_{11}$ | $[0.5, 1.5]^2$ | $\mathcal{L}_0 \cup \{x_2\}$ | $\mathbf{u}(\mathbf{x}) = x_1^{x_2}$ |
| $\Gamma_{12}$ | $[-1, 1]^2$ | $\mathcal{L}_0 \cup \{x_2\}$ | $\mathbf{u}(\mathbf{x}) = x_1^4 - x_1^3 + \frac{1}{2}x_2^2 - x_2$ |

### E.2. Benchmarks on Poisson's Equation

To compare SSDE with methods that apply symbolic regression (SR) directly to numerical solutions of differential equations, we constructed a dataset inspired by the symbolic regression task in the Nguyen benchmark (Uy et al., 2011). Specifically, we selected 12 target expressions from the Nguyen benchmark and used them as closed-form solutions to derive corresponding Poisson equations. The complete configuration of this dataset is summarized in Table 9. It is worth noting that expressions involving only a single variable result in ordinary differential equations (ODEs), while those involving two variables give rise to partial differential equations (PDEs). This setup allows us to evaluate the performance of SSDE and other baselines on both ODE and PDE settings in a controlled, ground-truth-aware environment. Leveraging its recursive exploration strategy, SSDE successfully recovers the Nguyen-12 expressions as exact closed-form solutions, whereas conventional symbolic regression methods like DSR (Petersen et al., 2020) and DSO (Mundhenk et al., 2021) are unable to fully reconstruct the target forms.

## F. Details of Experimental Results

SSDE successfully recovered the symbolic solutions for all tested partial differential equations, with the intermediate regressed expressions presented in Table 10. In contrast, the solutions obtained by PINN+DSR often omitted essential variables. We hypothesize that DSR's failure to identify correct solutions stems from two key limitations: (1) the increased dimensionality significantly expands the search space, making exploration more difficult; and (2) the policy gradients corresponding to different variables may interfere with each other during training, hindering convergence to the correct expression. Although the KAN method can approximate numerical solutions of PDEs with high accuracy through its network architecture, its symbolic regression capability remains limited. The resulting expressions tend to be dense and lack interpretability due to insufficient sparsity. In terms of efficiency, PR-GPSR performs poorly: in our experiments, it required approximately one day to complete 1500 generations. To ensure a fair comparison across methods, we capped the number of generations for PR-GPSR at 1500. Even within this limit, PR-GPSR consistently failed to recover the correct symbolic solutions across all benchmark equations.

*Table 10.* The stepwise regressed solutions of SSDE used to solve PDEs

| STEP | POISSON2D | POISSON3D | HEAT2D | HEAT3D | WAVE2D | WAVE3D |
|---|---|---|---|---|---|---|
| **STEP 1** | $x_1^2(2.5000x_1^2-1.3x_1)+\alpha_1$ | $2.5000x_1^4+\alpha_1$ | $0.5001t^2+\alpha_1$ | $\alpha_1-1.7000t-1$ | $\alpha_1 e^{-0.5002t}$ | $\exp(-\alpha_1-0.4999t)$ |
| **STEP 2** | $x_1^2(2.5000x_1^2-1.3x_1)+0.4997x_2(x_2-1.4005)-x_2$ | $2.5000x_1^4+0.6502\alpha_2-1.3003x_2^3-1.6502x_2$ | $0.5001t^2+\log(\exp(\alpha_2+2.4995x_1^4))$ | $\alpha_2-1.7000t+2.5000x_1^4-1$ | $\alpha_2\cos(1)\cdot e^{x_1^2-0.5002t}$ | $\exp(\alpha_2+1.0010x_1^2-0.4999t)$ |
| **STEP 3** | $x_1^2(2.5000x_1^2-1.3x_1)+0.4999x_2(x_2-1.4002)-x_2$ | $2.5000x_1^4-1.3003x_2^3+0.5000x_3^3-0.0003$ | $0.5001t^2+\log(\exp(2.4995x_1^4-x_2^2(1.3000x_2-0.0008))$ | $\alpha_3-1.7000t+2.5000x_1^4-1.3000x_2^3-1$ | $1.8508\cos(1)\cdot\sin(x_2)e^{x_1^2-0.5002t}$ | $\exp(1.0010x_1^2-0.4999t)\cdot\exp(\alpha_3+1.0010\ln(\cos(x_2)))$ |
| **STEP 4** | – | $2.5000x_1^4-1.3000x_2^3+0.4999x_3^2+2.3763e-5$ | $0.5000t^2+\log(\exp(2.5000x_1^4-x_2^2(1.3000x_2-8.1811e-7)))$ | $0.4998x_3^2-1.7000t+2.5000x_1^4-1.3000x_2^3-0.0006$ | $1.0000\sin(x_2)\cdot e^{x_1^2-0.5000t}$ | $\exp(1.0010x_1^2-0.4999t)\cdot\exp(x_3^2+1.0010\ln(\cos(x_2)))$ |
| **STEP 5** | – | – | – | $-1.7003t+2.5000x_1^4-1.3000x_2^3+0.4999x_3^2+0.0002$ | – | $\exp(x_3^2+1.0001x_1^2-0.5000t)\cdot\cos(x_2)$ |

