# OpenReview forum: "Closed-form Solutions: A New Perspective on Solving Differential Equations"
_ICML.cc/2025/Conference — ICML 2025 poster_

### Official Review · Reviewer_WYoN · 2025-02-25

**Overall Recommendation:** 3

**Summary:**

This work presents an approach to discover analytical solutions of PDEs using reinforcement learning. The authors combine several concepts in this work, including an optimization method constrained by IC/BCs, an iterative construction approach which generates solution skeletons and subsequently refines parameters of this expression, and finally an approach to find solutions along each dimension such that the approach can be extended to PDEs in multiple spatial dimensions. The general algorithm, which is visualized nicely in Figure 1, generates an expression from an RNN. Points from the domain are subsequently sampled and loss with respect to the ICs, BCs, and internal domain is calculated. Finally, a reward function is used to update the RNN in a reinforcement learning approach.

The authors present several experiments on ODEs, elliptic, and parabolic PDEs. Compared to the baselines, the proposed approach shows superior performance in most cases.

**Claims And Evidence:**

The authors present an ablation which supports their claim that the RSCO policy improves recovery rate and performance. As shown in Figure 2, the recovery rate is greatly diminished when RSCO is not used, and even moreso when recursive exploration is not used. The authors also claim that this approach will result in great improvements in accuracy and efficiency (4x speed up), but I do not see any experiments which support this claim. I would kindly request the authors to clarify this, or add the experimental results which support this claim.

The authors also present several tables with the closed form expressions which closely match the true solution; however, the claim that these are *novel* closed-form solutions is questionable. There are many approaches which have already been able to show the ability to learn closed-form solutions of ODEs and PDEs, including "Solving differential equations with genetic programming" (Tsoulos et al. 2006), and other approaches in Genetic Programming, Ant Colony Programming, etc. These approaches have even been shown to work for high dimensional (2 and 3 dimensional) PDEs.

Finally, the authors claim the ability to efficiently discover solutions for *complex* equations. However, I would argue that the DEs considered in this work are relatively simple. More challenging baselines, such as compressible Euler or Shallow Water Equations would be necessary to explore the performance of SSDE for more challenging equations. Likewise, the solutions of all DEs are polynomial. I see in the solution to the Heat2D (table 3), that SSDE identifies a solution which takes the log of an exponential. These effectively cancel out, but suggest the ability to find non-polynomial solutions. Solutions with sin and cos terms are also shown in the Gamma dataset.  Nonetheless, this suggests the approach may have challenges in finding more complex solutions. I believe this claim merits additional experiments to be supported.

**Essential References Not Discussed:**

The authors discuss genetic programming, providing a reference to work by Oh et al., but the foundational work in this field for DEs was introduced in 2006, "Solving differential equations with genetic programming," by Tsoulos and Lagaris.

**Experimental Designs Or Analyses:**

The design of the proposed experiments is sound, but more challenging equations and stronger baselines must be considered, as previously mentioned.

In terms of evaluation, I would suggest the use of a relative L1 or L2 error, in addition to the physics loss. This can be provided in the appendix, but it would be helpful to understand the performance of the approaches.

**Methods And Evaluation Criteria:**

The differential equations tested in this work make sense, but as mentioned in my previous comment, more challenging baselines should also be explored.

The evaluation of the baselines methods seems questionable. In their work, PR-GSPR is able to recover the analytical solutions for Poisson's equation, which is also tested in this work. Nonetheless, the results in Table 2 show that it failed to recover the correct solution across the experiments. Furthermore, the appendix states that hyperparameter exploration/tuning was not used for this approach, while a fairly large sweep was used for SSDE. A sweep should also be performed for the baselines, where applicable.

PINN+DSR also seems to be a weak benchmark, as one first solves the DEs with PINNs, and then fits an analytical solution to this, both of which will introduce some error. Of course, PINNs can be extremely sensitive to initialization and difficult to optimize, also introducing non-physical dynamics which might be very difficult to model with DSR. I believe simply providing the error of the PINNs with respect to the analytical solution would serve to better illustrate the performance of the method. As this work seems very similar to DSR, I believe this should still be considered as a baseline, however, it should be applied to data from the true solution.

The KAN baseline is interesting, but I believe this may be quite a bit outside of the realm in which KANs are usually used; however, I will admit I am not as familiar with this work.

I believe established analytical solvers, such as Eureqa and Mathetmatica would serve as stronger baselines in these experiments.

**Other Comments Or Suggestions:**

There are minor grammatical mistakes and typos throughout the work. Fixing these would improve reading.

Line 356, right column, a reference to comparison between SSDE and the numerical solution is said to exist in Appendix B, but this section only contains computing and hyperparameter details.

I believe the relationship of this work to the DSR paper should be discussed in more detail. There is some discussion in the appendix, but this is limited.

**Other Strengths And Weaknesses:**

Strengths
1. the authors design closed-loop extension of DSR which is able to be applied effectively to DEs
2. several elements of the proposed architecture improve the efficiency of the approach
3. the approach can be extended to arbitrarily-high-dimensional PDEs

Weaknesses
1. the paper lacks a limitations section, which is critical for clarity and transparency on the work
2. more difficult baseline experiments should be used to understand how well this approach can generalize to PDEs.
3. baseline models are weak, and not fairly optimized for the test cases
4. claims regarding efficiency are not supported experimentally: run times and number iterations are not provided

**Questions For Authors:**

1. Does the symbol library need to be defined a priori? For example, in the Gamma dataset, there are several solutions which contain sin and cos. Is it necessary that this be prescribed before finding the solution to the PDE?

2. Can knowledge be transferred between systems of PDEs, or is it necessary to train the model from scratch for each set of PDEs?

3. Table 11 in the appendix shows that the PINN+DSR and SSDE both obtain the correct analytical solution for $\Gamma_1$, yet SSDE has an MSE 22 orders of magnitude better than PINN+DSR. Is this due to small numerical errors in the constants of the solution which aren't shown in the provided solution, or is there some other reason?

4. What is the sensitivity to the IC and BC loss terms? Is the approach sensitive to hyperparameter tuning here?

5. The paper mentions exponential growth in the search space for an increase in the number of symbols. Does this limit the application of this approach to find solutions which have many terms?

I would like to thank the authors for their time, and I would gladly raise my evaluation if additional experimental results are provided to address my questions and concerns.

**Relation To Broader Scientific Literature:**

The key contribution to the literature is the extension of deep symbolic regression (DSR) to PDEs. The approach outlined in this work is fundamentally similar to DSR, using RNNs to generate expression trees, whose terms are taken from a defined library. Likewise, both approaches use reinforcement learning with additional risk seeking behavior to learn how an optimal policy for generating symbolic expressions.

More broadly, the paper focuses strongly on the performance improvements which come from risk-seeking optimization. Work in this field has recently been explored, referenced in the paper (see also arxiv 2302.09339). This work helps to illustrate the usefulness of risk-seeking optimization in RL.

**Theoretical Claims:**

Not applicable, as proofs are not included.

---

> ### Author Rebuttal · Authors · 2025-04-01
>
> Thank you for the feedback and suggestions, we will add clarification where needed and include suggestions as space permits.  Extra results: [here](https://anonymous.4open.science/r/SSDE-A47B/SSDE_icml2025_rebuttal.pdf). All figures/tables are from this link.
>
> **Q1: Add the experiments to support the efficiency improvement brought by RSCO.**
>
> Added ablation experiments of RSCO are perfomed on the Gamma benchmarks with single variable as shown in Fig 1. We conducted 20 experiments on each benchmark and counted the average time required to successfully find an analytical solution and the average number of explored expressions. Despite the RSCO method requires a slight increase in the number of exploration expressions, it greatly improves efficiency.
>
> **Q2: "Novel closed-form solutions".**
>
> This refers to SSDE’s parametric expression framework for high-dimensional PDEs (Appendix Table 10). RecurExp recursively constructs solutions, drastically shrinking search space and boosting efficiency.
>
> **Q3: Can SSDE discover solutions for more complex equations? Lacks of limitations section.**
>
> Added experiments on the nonlinear wave equation benchmarks are shown in Tabel 1 and Tabel 2. SSDE succeeds on nonlinear Wave2D/Wave3D benchmarks with non-polynomial solutions.
>
> We also acknowledge the limitations of SSDE in solving systems of DEs due to their inherent coupled-solution nature. Exploring the full solution space requires multiple RNNs, and evaluating reward-guided gradients for each solution necessitates novel methods. We aim to address this challenging issue in future work.
>
>
> **Q4: Discussion about baselines and evaluation method.**
>
> Thank you for your suggestions on baseline methods and evaluation metrics. First, we have added relative L2 error comparisons for baselines (Table 3). Regarding baselines:
>
> * We did not include PINN or DSR individually as baselines because our method focuses on discovering analytical solutions, which differs fundamentally from numerical approximation (PINN) or symbolic regression (DSR).
> * PINN+DSR represents a natural approach for symbolic solution discovery (solving DEs numerically with PINN followed by symbolic fitting via DSR), hence its inclusion as a baseline. For reference, we report PINN’s convergence error relative to analytical solutions (Table 4) and DSR’s performance on ground-truth data (Table 5).
> * We use the KAN method as a baseline because they did do the work of solving the symbolic solution of the differential equation in the paper, but this task may indeed be beyond the capabilities of KAN.
> * We added Mathematica 14.1 and the Finite Expression Method (FEX) [1] as stronger baselines (Table 3). While KAN attempts symbolic solution discovery via spline-based activation functions, FEX directly uses symbolic neurons. Both still fail to recover exact closed-form solutions for our benchmarks.
>
> We performed parameter tuning for the PRGP baseline, which improved its performance across benchmarks. Although the GP method can generate highly complex expressions to achieve good performance on the Van der Pol benchmark, it fails to recover closed-form solutions in high-dimensional settings. You mentioned that Poisson's equation could be solved in their paper, but the sample they tested in their original paper was very simple. Although the samples we tested look like  polynomials, the symbolic expressions involved are very complex and DSR cannot effectively identify them too. Our approach still works very well on these high-dimensional problems.
> The performance of the PRGP method can also refer to [2].
>
> [1] Finite Expression Method for Solving High-Dimensional Partial Differential Equations
>
> [2] An interpretable approach to the solutions of high-dimensional partial differential equations.
>
> Answer for `Question For Authors`:
>
> Q1: Symbol libraries are predefined, as in RL/GP-based heuristic search algorithms.
>
> Q2: SSDE requires retraining per PDE system. But transferability is an interesting concern, the solution of different DEs may have very large differences.
>
> Q3: Yes, MSE difference stems from truncating constants to significant digits.
>
> Q4: SSDE’s sensitivity to IC/BC terms mirrors Mathematica’s behavior.
>
> Q5: As we replied to the reviewer ateq(Q3), symbol library size increases search complexity. `RecurExp` it is used to solve the problem of increased search space caused by increased variables.
>
> We will thoroughly check and fix grammatical and clerical errors in the final submission. Thank you for your constructive feedback and for indicating your openness to revising your evaluation. We have thoroughly addressed all your questions and concerns through additional experiments, which are now included in the rebuttal document here and highlighted in Tables 2/3/4/5,Figure1. These results directly address your specific requests and strengthen the robustness of our claims. We kindly ask that you reassess our submission in light of these additions.

---

> > ### Comment · Reviewer_WYoN · 2025-04-02
> >
> > I would like to thank the authors for their work to address my concerns. I think it is important to recognize that methods to find analytical solutions for challenging PDEs are somewhat nascent. I have personally explored other baselines in this line of work, and I have consistently found it difficult to extend such works to data which more closely mimics real-world problems. In light of the improvements this approach shows in finding solutions to problems such as the 3D wave, I will raise my score. I would encourage the authors to push the limits of this approach, demonstrating failure cases as well, so that readers may better understand the limits of this approach.

---

> > > ### Author Response · Authors · 2025-04-06
> > >
> > > Thank you for your constructive feedback and recognition of our work's potential in this emerging field. We appreciate your suggestion to explore failure cases and have provided additional analysis in [this supplement](https://anonymous.4open.science/r/SSDE-A47B/SSDE_icml2025_rebuttal.pdf), including:
> > > - Quantitative comparisons between SSDE's symbolic solutions and numerical ground truth for Van der Pol (Figure 3)
> > > - Performance on the analytically intractable Wave3D* system (Tables 7-8), where SSDE maintains **boundary condition consistency** and **spatial distribution alignment** with numerical solutions (Figure 4), unlike PR-GPSR's no interpretability expressions.
> > >
> > > We fully agree with your observation about current methods' limitations. While baseline approaches often produce numerical-approximation-like solutions through over-parameterization, they sacrifice interpretability. Our analysis of the solution discrepancies (Figure3) indicates that SSDE's approximation boundaries on Wave3D* could plausibly arise from terms resembling
> > > $x^{2.5}$ in the latent solution – expressions undefined at the origin (x=0) and outside our current symbolic library's domain definitions. This appears to expose a fundamental limitation in handling such singularities, while pointing to targeted operator library expansion (e.g., fractional exponents with domain constraints) as a critical enhancement pathway.
> > >
> > > Regarding computational efficiency, we acknowledge SSDE's current time costs compared to PINNs. As you astutely note, integrating pretrained paradigms like [1] could bridge this gap – a promising direction for future work that would build on our demonstrated success in **scaling RL-based symbolic exploration to high-dimensional PDEs**.
> > >
> > > We hope these clarifications demonstrate our method's unique value in balancing interpretability with physical plausibility. Given the novel characteristics shown in 3D PDE systems and our thorough failure mode analysis, we would be grateful for further consideration of a score improvement to help advance this critical research direction.

---

### Official Review · Reviewer_vQDg · 2025-03-10

**Overall Recommendation:** 3

**Summary:**

The paper proposes a deep learning approach to obtain closed-form solutions for PDEs. The authors exploit this task through a Markov decision process and introduce an RL-based methodology. They also address acceleration and multi-dimensional problems, presenting an ablation study to support the proposed methods. While the proposed approach presents promising results, its applicability to general PDEs remains somewhat uncertain.

**Claims And Evidence:**

The advantages of treating the closed-form solution of the PDE as an RL problem should be articulated more clearly.
Based on the experimental results, it seems that the proposed approach demonstrates superior performance compared to existing methods, even in the absence of RSCO or recursive exploration. I would appreciate it if you could further elaborate on why you believe the proposed framework is particularly advantageous for handling symbolic expressions. Additionally, since RSCO was introduced with the intention of accelerating convergence, why the lack of RSCO results in a decrease in performance, as shown in the results presented in Figure 2?

**Essential References Not Discussed:**

No essential related works are missing.

**Experimental Designs Or Analyses:**

The proposed methodology demonstrates superior performance compared to existing approaches. However, the incorporation of policy gradients, RSCO, and recursive exploration likely increases its complexity. To assess this, I recommend comparing computational time and memory usage between the proposed methodology and baseline methods across 1D, 2D, and 3D cases.

**Methods And Evaluation Criteria:**

The paper currently focuses on smooth solutions, but most general PDEs do not possess classical solutions. As a result, generalized concepts of solutions, such as weak or distributional solutions, are considered, which are generally not unique. For the proposed approach to be of broader significance, it could be applicable/extendable to such scenarios. Could you elaborate on how the methodology might be extended to handle such cases? I am not requesting the addition of experiments but rather would appreciate a discussion on the potential for extending the proposed methodology to such practical scenarios.

**Other Comments Or Suggestions:**

There are some minor typos, for example:

* Line 153, first column: '$R\times\times\cdots\times R^n$' should be '$R\times R^n\times\cdots\times $'.

* Line 204, second column: 'We also note that the skeleton satisfy deterministic' should be 'We also note that the skeleton satisfies deterministic.'

**Other Strengths And Weaknesses:**

Please refer to the comments provided above.

**Questions For Authors:**

Please refer to the comments provided above.

**Relation To Broader Scientific Literature:**

If the closed form of the solution can be obtained, it would not only enhance the interpretability of solutions predicted by deep learning models but also provide valuable insights into the mathematical understanding of PDEs. This task effectively leverages the strengths of deep learning.

**Theoretical Claims:**

The paper does not include any theoretical content.

---

> ### Author Rebuttal · Authors · 2025-04-01
>
> Thank you for the feedback and suggestions, we will add clarification where needed and include suggestions as space permits.  Extra results: [here](https://anonymous.4open.science/r/SSDE-A47B/SSDE_icml2025_rebuttal.pdf). All figures/tables are from this link.
>
> **Q1: Advantages of Formulating PDE Solutions as an RL Problem.**
>
> The RL framework is critical for symbolic expression discovery because:
>
> * Symbolic Expressions Lack Gradient Backpropagation: Unlike PINN’s numerical solutions (which rely on gradient backpropagation), symbolic expressions require gradient-free optimization. Reinforcement learning (RL) provides policy gradients guided by reward signals, enabling direct exploration of interpretable expressions.
> * Superior Trade-off Between Complexity and Interpretability: Genetic programming (GP) methods optimize solutions via fitness functions but often overfit by increasing expression length and adding excessive constants. In contrast, our RL framework balances accuracy and simplicity by directly rewarding concise, interpretable expressions (e.g., avoiding unnecessary terms).
>
> Regarding RSCO’s impact:
>
> Figure 1 shows that RSCO reduces solution time despite a slight increase in explored expressions. Interestingly, under identical seeds, RSCO often finds solutions faster (e.g., mirroring hPINN’s boundary-condition prioritization for convergence acceleration).
>
> **Q2: Extending to Weak/Distributional Solutions**
>
> You raise an important point about generalizing to weak/distributional solutions. Potential extensions include:
>
> 1. Enriching the Symbol Library: Introducing neural operators or special functions (e.g., Bessel functions) to capture non-smooth solutions.
> 2. Combining Multiple Solutions: Treating SSDE’s outputs as particular solutions and aggregating results across multiple runs (analogous to classical DE solving techniques).
>
> We acknowledge this as a promising future direction and will explore it further.
>
> **Additional Clarifications**
>
> *  While we are unable to conduct a full run-time and memory-usage comparison due to time constraints, we can provide quantitative context for SSDE’s run-time: For the 3D heat equation, SSDE required ~2,700 seconds to converge. RSCO’s design ensures that memory usage remains comparable to DSR , as it avoids storing large amount of computation graphs for automatic differentiation.
> * We will thoroughly check and fix grammatical and clerical errors in the final submission.
> * The code and supplementary experiments are provided to clarify implementation details.
>
> We appreciate your feedback and welcome further discussion to address any remaining concerns.

---

> > ### Comment · Reviewer_vQDg · 2025-04-08
> >
> > Thank you for your detailed response and I appreciate for additional experiments. In particular, I believe that if the content of Q1 is clearly organized and included in the manuscript, it will greatly help in understanding the paper.

---

> > > ### Author Response · Authors · 2025-04-09
> > >
> > > Thank you for your constructive feedback. We will explicitly integrate the content of Q1 and expanded experimental results into the final manuscript to enhance readability. Given the novelty of our RL-based framework for symbolic solver– a critical step toward interpretable scientific machine learning – we would be deeply appreciative of further consideration for a score improvement to better reflect this work's potential in advancing the field.

---

### Official Review · Reviewer_ateq · 2025-03-14

**Overall Recommendation:** 1

**Summary:**

This paper proposes SSDE, a reinforcement learning-based framework for deriving closed-form symbolic solutions to differential equations. The authors introduce a risk-seeking constant optimization technique and recursive exploration strategy to enhance the method's efficiency. Experiments are conducted on various ordinary and partial differential equations to demonstrate the approach's effectiveness.

**Claims And Evidence:**

The paper claims that SSDE can effectively find closed-form solutions for differential equations without prior mathematical background. While the idea of combining symbolic learning with neural networks to solve PDEs is promising, the evidence provided is insufficient to fully support this claim. The experiments are limited to linear PDEs and ODEs, which can be solved using traditional methods.

**Essential References Not Discussed:**

Several important references are missing, including:

1. Works on neural operators (e.g., DeepONet [1], FNO [2]) (The algorithm presented in this study necessitates a dataset for both training and testing purposes. The nature of this task aligns with the fundamental concepts explored in neural operator research.)
2. Traditional numerical methods for PDEs [3]
3. Other reinforcement learning approaches for symbolic regression [4-9]


[1] Deeponet: Learning nonlinear operators for identifying differential equations based on the universal approximation theorem of operators
[2] Fourier neural operator for parametric partial differential equations
[3] Spline approximation, part 1: Basic methodology
[4] Finite Expression Method for Solving High-Dimensional Partial Differential Equations
[5] Deep Learning and Symbolic Regression for Discovering Parametric Equations
[6] Deep Reinforcement Learning-Based Symbolic Regression for PDE Discovery Using Spatio-Temporal Rewards
[7] Symbolic genetic algorithm for discovering open-form partial differential equations (SGA-PDE)
[8] Deep symbolic regression for physics guided by units constraints: toward the automated discovery of physical laws.
[9] Symbolic regression via neural-guided genetic programming population seeding

**Experimental Designs Or Analyses:**

The experimental design has several limitations:

1. Only linear PDEs and ODEs are tested
2. No comparison with traditional numerical methods
3. Limited parameter settings are explored
4. No analysis of computational efficiency
5. Dataset acquisition process is not explained

**Methods And Evaluation Criteria:**

The methodology combines reinforcement learning with symbolic regression and introduces novel optimization techniques. However, the evaluation criteria are limited to RMSE and recovery rate, which do not provide a comprehensive assessment of the method's capabilities compared to existing approaches.

**Other Comments Or Suggestions:**

1. Address mathematical notation inconsistencies and errors (e.g., extra multiplication sign on line 153, potential issue with F definition on line 153, missing 's' in "ordinary differential equation" on line 156, inconsistent use of subscripts for x on line 131)
2. Expand experimental validation to include nonlinear PDEs (e.g., NS equations, Gross-Pitaevskii equations)
3. Compare with traditional numerical methods and other machine learning approaches
4. Analyze computational efficiency and compare with traditional methods (e.g., spline interpolation)
5. Provide source code and implementation details
6. Clarify dataset acquisition process
7. Discuss the method's limitations and practical applications more thoroughly
8. Provide GPU details in the appendix (line 297)

**Other Strengths And Weaknesses:**

Strengths:

1. Novel combination of reinforcement learning and symbolic regression
2. Clear motivation and problem formulation
3. Introduction of risk-seeking constant optimization and recursive exploration techniques

Weaknesses:

1. Limited experimental validation (only linear PDEs and ODEs)
2. Lack of comparison with existing methods
3. No analysis of computational efficiency
4. Mathematical notation inconsistencies and errors
5. Limited discussion of practical applications
6. Missing supplementary material and source code

**Questions For Authors:**

1. How does SSDE compare to other reinforcement learning approaches for symbolic regression [2-5]?
2. Can SSDE handle nonlinear PDEs like NS, Gross-Pitaevskii equations?
3. What is the computational efficiency compared to traditional methods [1]?
4. How was the training dataset acquired?
5. What are the limitations of the proposed approach?
6. How does the method scale with problem size and desired accuracy?
7. Can the algorithm be parallelized on CPUs and GPUs?

[1] Spline approximation, part 1: Basic methodology
[2] Finite Expression Method for Solving High-Dimensional Partial Differential Equations
[3] Deep Learning and Symbolic Regression for Discovering Parametric Equations
[4] Deep Reinforcement Learning-Based Symbolic Regression for PDE Discovery Using Spatio-Temporal Rewards
[5] Symbolic genetic algorithm for discovering open-form partial differential equations (SGA-PDE)
[6] Deep symbolic regression for physics guided by units constraints: toward the automated discovery of physical laws.
[7] Symbolic regression via neural-guided genetic programming population seeding

**Relation To Broader Scientific Literature:**

The paper does not adequately situate itself within the broader literature. Key related works in neural operators, traditional numerical methods, and other symbolic regression approaches are not sufficiently discussed or compared.

**Theoretical Claims:**

The theoretical claims about the method's ability to handle high-dimensional PDEs are not sufficiently supported. The paper lacks a rigorous theoretical analysis of the algorithm's convergence properties and computational complexity.

---

> ### Author Rebuttal · Authors · 2025-03-27
>
> Thank you for the feedback and suggestions, we will add clarification where needed and include suggestions as space permits.  Extra results: [here](https://anonymous.4open.science/r/SSDE-A47B/SSDE_icml2025_rebuttal.pdf). All figures/tables are from this link.
>
> **Q1: How does SSDE compare to other reinforcement learning approaches for symbolic regression [2-5]?**
>
> Added comparisons with [2] (`FEX` in Table 3), showing SSDE’s superiority. [2] will be cited in related work. We performed parameter tuning for the PRGP baseline, which improved its performance across benchmarks. Although the GP method can generate highly complex expressions to achieve good performance on the Van der Pol benchmark, it fails to recover closed-form solutions in high-dimensional settings. [3-5] focus on differential equation discovery via symbolic regression, addressing different problem settings and thus not directly comparable.
>
> **Q2:What is the computational efficiency compared to traditional methods(Spline approximation)? Can SSDE handle nonlinear PDEs?**
>
> SSDE targets closed-form solutions (unattainable by numerical methods like [1]), offering interpretability and mesh-free advantages. Traditional methods trade efficiency for precision via mesh refinement (non-convergence risks).
>
> We supplement the experiments on nonlinear wave equations. The experimental results is shown in Table 2 and Table 3, which suggests that our method is competitive. Beside, we add Mathematica 14.1 (state-of-the-art symbolic solver) as the extra baseline (In Table 3).
>
>
> **Q3: Lacks convergence/complexity analysis. How does scalability depend on problem size and accuracy?**
>
> 1. Convergence follows risk-seeking policy gradients[1].
> 2. The search complexity increases exponentially with the size of the library, but the RecurExp method significantly reduces the complexity caused by the increase in variables(Section 4.3). More detailed analysis will be added in the final submission. Accuracy depends on symbolic expressivity, bypassing numerical discretization limits.
>
> [1] Deep symbolic regression: Recovering mathematical expressions from data via risk-seeking policy gradients.
>
>
> **Q4:How was the training dataset acquired?Can SSDE be parallelized on CPUs and GPUs?**
>
> SSDE uses a **self-supervised paradigm** without external datasets. Collocation points are sampled randomly during training. RNN is used to generate candidate expressions evaluated via PDE residual loss and BC/ICs penalties. Policy gradients are updated to ensure identified solutions are constrained by the governing equations. It supports CPU/GPU parallelization, but experiments were conducted on CPUs due to the lack of access to GPU resources.
>
> **Q5:What are the limitations of SSDE?**
>
> SSDE currently lacks efficiency in solving systems of DEs due to their inherent coupled-solution nature. Exploring the full solution space requires multiple RNNs, and evaluating reward-guided gradients for each solution necessitates novel methods. We aim to address this challenging issue in future work.
>
> **Other suggestions**
>
> 1. The evaluation criteria are limited to RMSE and recovery rate, which do not provide a comprehensive assessment of the method's capabilities compared to existing approaches.
>
>    Added L2 relative error comparisons(Table 3). While recovery rate and MRMSE are tailored to our problem's unique requirements, we clarify their rationale:
>
>    Recovery rate: Assesses closed-form solutions by requiring both symbolic and numerical equivalence (<1e-8 MSE threshold).
>
>    MRMSE: Quantifies equation satisfaction when symbolic ground truth is unavailable, complementing traditional numerical metrics.
>
> 2. Address mathematical notation inconsistencies and errors.
>
>    We will thoroughly check and fix grammatical errors in the final submission. The F definition on line 153 refers to: Evans, L. C. Partial differential equations.
>
>
> 3. Limited parameter settings are explored.
>
>    We provide a full scan of the parameters in Appendix B.
>
> 4. Key related works in neural operators, traditional numerical methods, and other symbolic regression approaches are not sufficiently discussed or compared. Several important references are missing.
>
>    We have discussed neural operators, traditional numerical methods, and symbolic regression approaches in the introduction/related work, emphasizing their inability to achieve closed-form solutions for DEs.  While neural operator references are included (citing their published works), we acknowledge the need for deeper engagement with additional literature. We will supplement references in the final version.
>
> SSDE targets discovering closed-form solutions for DEs—a goal distinct from numerical approximation[1] or symbolic equation discovery (e.g., [3-5]). Given clarifications on SSDE's focus and added experiments highlighting contributions, we kindly request reconsideration of your score per these revisions.

---

> > ### Comment · Reviewer_ateq · 2025-04-02
> >
> > 1. **Regarding Q1**:
> >    Thank you for the additional experimental results. However, likely due to space constraints, some key aspects remain unclear. I still have the following concerns:
> >    • **Algorithmic Differences**: Could you provide a more detailed technical explanation of how your method fundamentally differs from similar approaches in this category?
> >    • **Experimental Comparisons**: It appears that the comparisons I previously referenced ([6], [7]) were not addressed in your response.
> >
> > 2. **Regarding Q2**:
> >    I appreciate the supplementary experiments. However, based on the results in your linked materials, SSDE significantly underperforms the baseline PR-GPSR on the given nonlinear problem. Additionally, the van der Pol example is a single-variable ODE, which is relatively simple. To better assess the method’s robustness, could you evaluate it **on multivariate nonlinear PDEs without analytical solutions**, such as: 3D Navier-Stokes equations ; 3D Gross-Pitaevskii equations.
> >
> > 3. **Regarding Q3**:
> >    Could you experimentally demonstrate how computational overhead scales as the library size increases?
> >
> > If these concerns are adequately addressed, I would be happy to reconsider my evaluation score.

---

> > > ### Author Response · Authors · 2025-04-06
> > >
> > > Thank you for your constructive feedback. Extra results:[here](https://anonymous.4open.science/r/SSDE-A47B/SSDE_icml2025_rebuttal.pdf).
> > >
> > > Regarding Q1:
> > >
> > > Since SSDE is not designed for symbolic regression tasks and its objectives and application scenarios differ from those of methods [6,7], we did not conduct additional comparative experiments. We categorize [2-7] into three classes:
> > > 1. Symbolic PDE Solvers (e.g., FEX [2]): No additional data set is required, relying only on the DEs, ICs and BCs to find the analytical solution of the differential equation.
> > >    - Key distinctions in algorithm:
> > >      - **Representation:** SSDE uses RNNs to generate serialized expression trees (preserving sequential semantics), while FEX employs DNNs to build weighted trees (losing interpretability).
> > >      - **Optimization:** SSDE proposes RSCO for fast constant screening vs FEX's gradient-based parameter tuning.
> > >      - **Search Strategy:** SSDE proposes recursive dimension-wise exploration, unlike FEX's node-wise growth.
> > >
> > >     In general, both SSDE and FEX regard solving the symbolic solution of differential equations as a combination optimization problem, and the advantage of SSDE lies in its efficient search space design and interpretability, while FEX tends to be a learnable symbolic network design like NAS[1].
> > >
> > >     [1] Neural Architecture Search with Reinforcement Learning.
> > >
> > > 2. Symbolic Regression (e.g., DSR variants [6,7]): The real data set is required to discover the relationship between variable x and label y in the data set.
> > >    - Key distinctions in algorithm:
> > >      - Reward design (physics-driven residuals vs data-driven MSE)
> > >      - Exploration strategy (recursive decomposition vs genetic programming used in [7])
> > >      - Constant optimization (RSCO vs Local constant optimization)
> > >      - Application scope (PDE solving vs general regression)
> > >
> > > 3. PDE Discovery Methods ([3-5]): These require observational data to identify the governing PDEs underlying the system, focusing on data-driven discovery of differential operators rather than solving known equations, representing an orthogonal problem setup.
> > >
> > > Regarding Q2:
> > >
> > > We address your concerns with new evidence:
> > > 1. Interpretability Advantage: While PR-GPSR achieves lower residual on Van der Pol , SSDE attains practically sufficient accuracy (relative l2 error < 0.005) with solutions that are 10× more parameter-efficient and inherently interpretable (Figure 3).
> > > 2. Wave3D* Benchmark (Multivariate PDE without analytical solutions, Table 7).
> > >    * Our time-constrained comparison with top baselines (Table 8) reveals SSDE’s consistent discovery of simpler expressions while preserving spatiotemporal fidelity through recursive exploration. Figure 4 confirms that SSDE’s solutions align quantitatively with finite-difference numerical results, with RSCO ensuring strict boundary compliance – a critical advantage for physical modeling.
> > >
> > >    * SSDE's approximation limitations stem from intrinsic spatial complexity, chanllenging to approximate the solution space with limited symbols.  In contrast, while PR-GPSR produces more complex expressions, its solution exhibits significant boundary condition discrepancies and loses interpretability.
> > >
> > >     These results underscore SSDE's unique capability in balancing solution simplicity with physical plausibility in high-dimensional PDE systems.
> > >
> > > Regarding Q3:
> > >
> > > We experimentally demonstrate the scaling of computational overhead with increasing library size by analyzing the impact on both convergence time to the analytical solution and the number of explored expressions across the Gamma 1-4 benchmarks (as shown in Figure 2). The base symbol library includes only the variables required for the analytical solution, while extended configurations incorporate additional mathematical functions ($\sin$, $\cos$, $\exp$, $\log$) that are absent from the target analytical expressions themselves.
> > >
> > >
> > > We appreciate your consideration and stand ready to provide additional clarifications.

---

### Decision · Program_Chairs · 2025-05-01

**Decision:**

Accept (poster)

**Comment:**

This submission presents a reinforcement learning based approach for solving differential equations symbolically. The proposed method is quite novel and is used in experiments on ODEs, elliptic and parabolic PDEs, and eventually also the 3d wave equation. The improvement over the baselines is clear, although there was some disagreement among reviewers about the sufficiency of the experiments considered. In fact, 2/3 reviewers urged the authors to consider more realistic, challenging PDEs such as those coming from physics; the authors have agreed and presented some more experiments in this direction. The authors addressed the majority of these concerns in the rebuttal period and it is my opinion that the paper should be accepted despite the fact that one reviewer recommends rejection.